# Identification and initial characterization of Hfq-associated sRNAs in *Histophilus somni* strain 2336

Bindu Subhadra[1], Dianjun Cao[1], Roderick Jensen[2], Clayton Caswell[3], Thomas J. Inzana [1,3] *

1 College of Veterinary Medicine, Long Island University, Brookville, New York, United States of America,
2 College of Science, Virginia Tech, Blacksburg, VA, United States of America, 3 Virginia-Maryland College of Veterinary Medicine, Virginia Tech, Blacksburg, VA, United States of America

* Thomas.Inzana@liu.edu

**Data Availability Statement:** The complete RNA sequences are accessible through NCBI BioProject

## Abstract

Small RNAs (sRNA), in association with the global chaperone regulator Hfq, positively or negatively regulate gene expression in bacteria. For this study, *Histophilus somni* sRNAs that bind to Hfq were identified and then partially characterized. The Hfq-associated sRNAs in *H. somni* were isolated and identified by co-immunoprecipitation using anti-Hfq antibody, followed by sRNA sequencing. Sequence analysis of the sRNA samples identified 100 putative sRNAs, out of which 16 were present in pathogenic strain 2336, but not in non-pathogenic strain 129Pt. Bioinformatic analyses suggested that the sRNAs HS9, HS79, and HS97 could bind to many genes putatively involved in virulence/biofilm formation. Furthermore, multi-sequence alignment of the sRNA regions in the genome revealed that HS9 and HS97 could interact with sigma 54, which is a transcription factor linked to important bacterial traits, including motility, virulence, and biofilm formation. Northern blotting was used to determine the approximate size, abundance and any processing events attributed to the sRNAs. Selected sRNA candidates were confirmed to bind Hfq, as determined by electrophoretic mobility shift assays using sRNAs synthesized by *in vitro* transcription and recombinant Hfq. The exact transcriptional start site of the sRNA candidates was determined by RNA ligase-mediated rapid amplification of cDNA ends, followed by cloning and sequencing. This is the first investigation of *H. somni* sRNAs that show they may have important regulatory roles in virulence and biofilm formation.

## Introduction

*Histophilus somni* is one of the major etiologic agents responsible for bovine respiratory disease (BRD) complex, and is also responsible for multiple systemic diseases, including septicemia, thrombotic meningoencephalitis, myocarditis, arthritis, infertility, and abortion [1–5]. BRD is the result of complex interactions between *H. somni*, the environment, host, and other bacterial and viral pathogens [6]. Genome sequences, complete or partial, have been obtained for over 65 *H. somni* strains (NCBI, GenBank), which have enabled whole genome

under accession number PRJNA845786 (https://www.ncbi.nlm.nih.gov/bioproject/PRJNA845786).

**Funding:** This work was supported by USDA-NIFA grant 2017-67015-26797 to TJI, and funds from Long Island University. The funders had no role in study design, data collection and analysis, decision to publish, or preparation of the manuscript.

**Competing interests:** The authors have declared that no competing interests exist.

comparisons to be made that provide improved understanding of the differences and similarities in the number and composition of genes involved in metabolism, virulence, and pathogenesis [7]. *H. somni* strain 2336, a respiratory disease isolate, has a genome of 2.2 Mbp harboring 2044 predicted open reading frames (ORFs), of which only 1569 have an assigned biological function [8].

Bacterial small RNAs (sRNAs) are a class of regulatory RNAs typically ranging in size from 40 to 500 nucleotides [9]. sRNAs function primarily as post-transcriptional regulators by binding to specific mRNA targets and altering their stability and/or translation, or by interacting with proteins and modulating their activity [10]. The sRNA interactions are often mediated by RNA chaperones such as Hfq, which assist in annealing and stabilizing sRNA and the sRNA-target complex [11]. Functional characterization of several known bacterial sRNAs indicated that they contribute to post-transcriptional regulation, which plays crucial roles in a variety of biological processes including expression of outer membrane proteins [12], iron homeostasis [13], quorum sensing [14], and bacterial virulence [15]. According to the sRNA database sRNAMap, more than 900 sRNAs have been identified thus far in microbial genomes, of which most are transcribed from the intergenic regions [9]. In *Escherichia coli*, more than 100 sRNA regulators have been reported, of which the majority act in association with Hfq [16]. In recent years, sRNAs have been identified in many other bacterial species including *Vibrio cholerae*, *Pseudomonas aeruginosa*, *Staphylococcus aureus*, *Streptomyces coelicolor*, *Salmonella enterica*, *Mycobacterium tuberculosis*, *Bacillus subtilis*, *Listeria monocytogenes*, and others [17–23]. The sRNAs found in genomes of bacterial pathogens are reported to control various housekeeping activities and functions required for pathogenesis [24].

It is difficult to identify all functional elements encoded for in a bacterial genome, particularly sRNAs, with the current genome annotation programs. Several characteristics of sRNAs make them difficult to identify by straightforward computational and experimental approaches [25]. Thus, very little is known about the non-coding RNAs in the bovine pathogen *H. somni*. An RNA sequencing-based experimental annotation of the *H. somni* 2336 genome lead to the identification of 94 small RNA candidates [8]. In this study, we isolated sRNAs that bound to an epitope-tagged *H. somni* Hfq protein, sequenced, and partially characterized these Hfq-bound RNAs. Bioinformatic- and laboratory-based analysis of these sRNAs predicted that many of the sRNAs are involved in virulence and biofilm formation in *H. somni*. This is the first report that describes the identification and preliminary characterization of novel Hfq-associated sRNA candidates using the described approach to discover non-coding sRNAs in *H. somni*.

## Materials and methods

### Bacterial strains and culture conditions

The bacterial strains and plasmids used in this study are shown in Table 1. *H. somni* strain 2336 was cultivated on Columbia agar (BD Difco™, Franklin Lakes, NJ) supplemented with 5% sheep blood, and incubated in 5% $CO_2$ at 37˚C overnight. Planktonic cultures of *H. somni* 2336 were cultured in Columbia broth (BD Difco™) supplemented with 0.1% Trizma base and 0.01% thiamine monophosphate [26] (CTT), and incubated at 37˚C with shaking at 180 rpm. *E. coli* was cultivated in Lysogeny Broth (LB) medium (BD Difco™) at 37˚C with ampicillin (100 μg/ml) added for growth of recombinant strains.

### DNA manipulations

Standard cloning procedures were carried out as described [27]. Oligonucleotides were purchased from Integrated DNA Technologies (IDT, Research Triangle Park, NC) and are shown in Table 1. The genomic DNA from *H. somni* 2336 and the plasmid DNA from *E. coli* were

**Table 1. Bacterial strains, plasmids, and oligonucleotides used in this study.**

| Strain or plasmid or oligonucleotide | Relevant characteristics | Reference or source |
|---|---|---|
| Strains | | |
| *H. somni* strain 2336 | Pneumonia isolate | [5] |
| *E. coli* One Shot™ Top 10 | F- *mcrA* Δ( *mrr-hsd*RMS-*mcr*BC) Φ80*lacZ*ΔM15 Δ *lac*X74 *rec*A1 *ara*D139 Δ( *araleu*)7697 *gal*U *gal*K *rps*L (StrR) *end*A1 *nup*G | Invitrogen, USA |
| *E. coli* One Shot™ BL21 Star™ (DE3) | F⁻*omp*T *hsd*S_B (r_B⁻, m_B⁻) *galdcmrne*131 (DE3) | Invitrogen, USA |
| Plasmid | | |
| pET101/DTopo | ColE1/pMB1/pBR322/pUC ori; *lacI*; multicloning sites; His-tag (6x) V5 epitope tags; Ampicillin resistance | Invitrogen, USA |
| pUC19 | pMB1 ori; 5' terminal *lacZ*; P_lac; Ampicillin resistance | Invitrogen, USA |
| Oligonucleotides | | |
| Hfq_overex_F | CACCATGGCAAAA GGACAATCACT | This study |
| Hfq_overex_R | TTCAGTTTGATTTTCAGCTAATACTTCCG | This study |
| HS9_in vitro transcript_F | TAATACGACTCACTATAGGGAATAATACTGAATAATAGCTATAT | This study |
| HS9_in vitro transcript_R | AGCAAAAAAAAGATGCCGTTA | This study |
| HS79_in vitro transcript_F | TAATACGACTCACTATAGGGGTGTATAATAGAAATCAAATAATC | This study |
| HS79_in vitro transcript_R | AAAAAATTTTTTTTGCGTGGATT | This study |
| HS97_in vitro transcript_F | TAATACGACTCACTATAGGGCGCTGGCAGGCTAATGAC | This study |
| HS97_in vitro transcript_R | GTCAAATAAAAAATAGGTCGTT | This study |
| HS9_Northern | AAGTAGCACTTGTGGCAGCTTACGCC | This study |
| HS79_Northern | GCTATAGGGCTTATCCGTTACAAGTG | This study |
| HS97_Northern | TATTAGTCATTAGCCTGCCAGCGTAC | This study |
| 5' RACE adapter | GCUGAUGGCGAUGAAUGAACACUGCGUUUGCUGGCUUUGAUGAAA | This study |
| 5' RACE outer primer | GCTGATGGCGATGAATGAACACTG | This study |
| 5' RACE inner primer | CGCGGATCCGAACACTGCGTTTGCTGGCTTTGATG | This study |
| HS79_RACE_gs outer primer | CCTGTCGGTAACTATCCCTAAGC | This study |
| HS79_RACE_gs inner primer | CCCAAGCTTTGGCTATAGGGCTTATCCGTTAC | This study |
| HS97_RACE_gs outer primer | CAATACTGCGGAAACGAGCTG | This study |
| HS97_RACE_gs inner primer | CCCAAGCTTCGACCTTCAATCCACCCAACTT | This study |

isolated using the MasterPure™ DNA purification kit (Biosearch Technologies, Middleton, WI) and Zyppy plasmid miniprep kit (Zymo Research, Irvine, CA), respectively. The plasmids were introduced into *E. coli* by heat shock [28]. Phusion® High-Fidelity DNA Polymerase (New England Biolabs (NEB), Ipswich, MA) was used for polymerase chain reaction (PCR) amplification of DNA fragments. The DNA fragments were eluted and purified from the gel using the Zymogen gel DNA recovery kit (Zymo Research). Restriction and DNA modifying enzymes were purchased from NEB. The nucleotide sequences of recombinant strains were confirmed by sequencing (Psomagen, Inc., Brooklyn, NY).

## Overexpression and purification of hexahistidyl-tagged Hfq (His6-Hfq)

For the over-expression of His$_6$-Hfq, the complete coding region of Hfq (291 bp) of *H. somni* 2336 was amplified by PCR using the primer pair Hfq_overex_F/ Hfq_overex_R, and subcloned into the His-tag expression vector pET101/DTopo yielding pET-*hfq*. The recombinant plasmid was introduced into *E. coli* One Shot™ BL21 Star™ (DE3) cells (Invitrogen, Carlsbad, CA) by transformation. The overexpression of Hfq was induced by adding 1 mM isopropyl β-D-1-thiogalactopyranoside (IPTG) at an OD$_{600}$ of 0.6, and incubation of the culture was continued for 5 h at 37°C. Induced cells were harvested by centrifugation (5000 x *g* for 15 min)

and lysed by sonication (QSonica Sonicators, Newtown, CT) in lysis buffer [50 mM $NaH_2PO_4$ (pH 8), 500 mM NaCl, and 10 mM imidazole]. After centrifugation, the soluble supernatant was collected, and the recombinant protein Hfq was purified using nickel-nitrilotriacetic acid-agarose (Qiagen, Germantown, MD), following the manufacturer's protocol. The purified Hfq was dialyzed overnight against phosphate buffered saline (PBS), pH 7.4, to remove imidazole and stored at -80˚C. The purification of over-expressed Hfq was confirmed by SDS-PAGE followed by Western blotting with horseradish peroxidase (HRP)-conjugated antibody (Sera-Care, Milford, MA) to the V5 epitope of the recombinant protein.

## Production of antibodies to Hfq

Three young adult female Sprague Dawley rats were injected subcutaneously with an emulsion containing 30 μg of purified recombinant Hfq mixed with 120 μl of PBS and 150 μl of Freund's complete adjuvant (Thermo Fisher Scientific). Before the injection, 300 μl of peripheral blood was collected from the tail vein to obtain pre-immune serum while the animals were restrained in a restraining tube. Three weeks later, 300 μl of blood was collected to determine the serum antibody titer, followed by a booster injection containing 30 μg of purified Hfq in Freund's incomplete adjuvant (Thermo Fisher Scientific), as described above. The animals were boosted a third time three weeks later with 100 μg of purified Hfq in TiterMax Gold adjuvant (Sigma Aldrich, St. Louis, MO). Three weeks after the third immunization, blood was collected from the rats, and an antibody titer of >1:10,000 to purified Hfq was confirmed by enzyme-linked immunosorbent assay (ELISA), as described [29]. A fourth booster injection of 300 μg of purified Hfq in PBS was given intravenously at this time. Three days after the final booster, the rats were euthanized with $CO_2$ and exsanguinated by cardiac puncture. The blood was allowed to clot, and the sera stored at -20˚C. All procedures on animals were approved by the Long Island University Institutional Animal Care and Use Committee under protocol ID number 2021–001, February 17, 2021.

## Isolation of Hfq-associated sRNA

Hfq-associated sRNAs were co-immunoprecipitated, as previously described [21]. Anti-Hfq IgG was isolated from the serum using the ImmunoPure® Melon™ gel IgG spin purification kit (Thermo Fisher Scientific), according to the manufacturer's instructions. After dialysis against coupling buffer, the purified IgG was coupled to PIERCE™ NHS-activated magnetic beads (Thermo Fisher Scientific), as recommended by the manufacturer, and stored at 4˚C until used. Hfq and associated sRNAs were co-precipitated from cell extracts of *H. somni* after growing the bacteria, as described above, to stationary phase. Following centrifugation (5000 x *g* for 15 min) the bacteria were washed in binding/washing buffer [50 mM Tris-HCl (pH 7.5), 150 mM NaCl, 0.05% Tween 20, and protease inhibitor (Thermo Fisher Scientific)] followed by sonication to lyse the bacteria (20% Amp, 5 sec bursts with 30 sec intervals on ice for a total of 30 min). The unbroken cells were removed by centrifugation at 14,500 X *g* for 45 min, and the supernatant was collected. The co-immunoprecipitation of Hfq and associated sRNAs were carried out according to the protocol given for PIERCE™ NHS-activated magnetic beads (Thermo Fisher Scientific). The Hfq-associated sRNAs were separated from Hfq using TRIzol™ LS reagent (Invitrogen) according to the manufacturer's protocol. To control for nonspecific binding of sRNAs to the magnetic beads, the immunoprecipitation process was carried out with *H. somni* cell extract and magnetic beads not coupled to anti-Hfq IgG.

## RNA extraction

Total RNA, including microRNA, were extracted from *H. somni* cells using the Qiagen RNeasy® Mini Kit (Qiagen) and *mir*Vana™ miRNA isolation kit (Life Technologies,

Carlsbad, CA), respectively, following the manufacturer's protocol. The quality and quantity of the RNAs were evaluated by NanoDrop (Thermo Fisher Scientific) and gel electrophoresis.

## RNA sequencing and data analysis

The RNA samples were processed for cDNA library construction and sequencing at the Center for Genomics and Bioinformatics, Indiana University, Bloomington, IN. The cDNA libraries were prepared using the TruSeq small RNA preparation kit (Illumina Baltimore, Baltimore, MD) and the sequencing was performed using Illumina NextSeq 75 cycles High Output kit (Illumina) following the manufacturer's protocol. Reads were adapter-trimmed and quality filtered using Trimmomatic 0.38. An average base quality score of 20 over a window of 3 bases was set as the cutoff threshold. Cleaned reads were mapped to the *H. somni* 2336 reference genome sequence retrieved from NCBI (NC_010519.1) using Geneious Prime® 2021.0.3 (http://www.geneious.com). Geneious Prime® 2021.0.3 was set to the default parameters to generated the read coverage and read counts per gene. All intergenic regions of >30 bp were considered for read count analysis. The promoter and rho-independent terminators of the intergenic regions were predicted by BPROM (Softberry Inc., Mt. Kisco, NY) and ARNold [30], respectively. The criteria used for the prediction of promoters and terminators is detailed in S1 File. sRNA secondary structures were predicted by RNAfold webserver with default parameters [31]. The complete RNA sequence for sample GSF2751-sRNA Hfq-8 is accessible through NCBI BioProject under accession number PRJNA845786 (https://www.ncbi.nlm.nih.gov/bioproject/PRJNA845786). GSF2751-sRNA Hfq-8 is one of six repetitive samples from the same strain.

## *In vitro* RNA transcription

The DNA sequences of the sRNAs HS9, HS79, and HS97 were amplified from *H. somni* genomic DNA by PCR using forward primers harboring a T7 promoter sequence at the 5'-end. The PCR products were used as template for the synthesis of sRNAs HS9, HS79, and HS97 using MAXIscript® Kit (Invitrogen), according to the manufacturer's instructions. The transcribed product was treated with DNAse to remove the DNA template, followed by purification of the RNA using RNeasy® MinElute® Cleanup Kit (Qiagen) according to the manufacturer's protocol. The specificity and quality of the transcribed RNAs were confirmed by TBE (Tris/Borate/Ethylenediaminetetraacetic acid)-Urea gel (15% acrylamide) electrophoresis.

## Northern blotting

Based on the sequence similarity to currently known sRNAs that affect virulence in other Gram-negative bacteria, three of the sRNA candidates (HS9, HS79, and HS97) were selected for further studies. Northern blotting of the sRNA candidates was conducted using North2-South® Chemiluminescent Hybridization and Detection Kit (Thermo Fischer Scientific) according to the manufacturer's protocol. Briefly, the RNA isolated from *H. somni* cells was loaded onto a TBE-Urea gel (15%) and run at a constant voltage of 180 V for 85 min until the bromophenol blue tracking dye reached the bottom of the gel. As a size marker, a low range ssRNA ladder (New England Biolabs, MA, USA) was also loaded. Following electrophoresis, the RNA gels were rinsed in five gel volumes of 0.5x TBE buffer for 30 min. RNA was then transferred onto nylon membranes (Biodyne™ Precut Nylon Membranes, Thermo Fisher Scientific) at 20 V for 90 min, using Trans-Blot SD Semi-Dry Transfer Cell (Biorad, Hercules, CA). The RNA was then cross-linked to the membrane by baking at 80°C for 15 min before the hybridization process. Pre-hybridization and hybridization procedures were carried out according to the manufacturer's protocol. For hybridization, denatured (boiled) 5'-

biotinylated probes (26 bp in length; 30 ng per ml) were added to the hybridization buffer and incubated with the membranes overnight with shaking at 42˚C. The membranes were washed and developed according to the manufacturer's protocol, followed by imaging with ChemiDoc Imaging System (Biorad).

## RNA ligase-mediated (RLM)-rapid amplification of cDNA ends (RACE)

RLM-RACE was performed using FirstChoice® RLM-RACE Kit (Thermo Fisher Scientific) according to the manufacturer's protocol. The PCR products were visualized by agarose gel electrophoresis, purified, cloned into pUC19, and the recombinant clones were sequenced to identify transcriptional start sites.

## Electrophoretic mobility shift assay (EMSA)

EMSA was carried out to confirm the interaction of the three sRNAs selected (HS9, HS79, and HS97) with recombinant Hfq. Two hundred ng of *in vitro* transcribed sRNAs and recombinant Hfq (varying concentrations of 0.65 μM– 5 μM) were added to 15 μl of reaction mixture buffer (consisting of 10 mM Tris-HCl, pH 7.4, 6% glycerol, 50 μg/ml BSA, and 0.75 μg Baker's yeast tRNA), and incubated at room temperature for 30 min. The reaction mixtures were electrophoresed on a pre-run (100 V for 20 min) Novex™ retardation gel (6% native polyacrylamide gel) (Thermo Fisher Scientific) in 0.5x TBE buffer at 100 V for 90 min followed by staining with GelGreen® nucleic acid gel stain (Biotium, Fremont, CA) for visualization.

## Results

### Enrichment of Hfq-associated transcripts by co-immunoprecipitation

*H. somni* 2336 cells were grown in CTT broth to early stationary phase because Hfq expression has been shown to be upregulated in stationary phase cultures [32]. The *H. somni* cell extract was incubated with magnetic beads coupled with anti-Hfq IgG to bind Hfq and the associated sRNAs, as described in Materials and Methods. The successful recovery and quality of the extracted sRNAs were confirmed by RNA gel electrophoresis, followed by analysis using the Agilent 4200 TapeStation system (Agilent Technologies, Inc., Santa Clara, CA). cDNA libraries were constructed from 6 independent sRNA preparations that had the greatest RNA concentration and least salt contamination, followed by deep sequencing them individually using Illumina technology. The results of the RNA sequencing are presented in Table 2.

### Mapping of identified RNA sequences

Illumina sequencing generated a combined total of 20 million reads from the cDNA libraries created from 6 sRNA samples. The size of the sequencing reads was adequate for mapping

**Table 2. RNA sequencing result summary.**

| Sample | Barcode sequence | PF Clusters | % Perfect Barcode | % PF Clusters | % > = Q30 Bases | Mean Quality Score | Mapped Reads |
|---|---|---|---|---|---|---|---|
| GSF2751-sRNA_Hfq-1 | CTATACAT | 4,416,420 | 100 | 100 | 88.01 | 32.85 | 633,747 |
| GSF2751-sRNA_Hfq-11 | TCGGCAAT | 2,667,639 | 100 | 100 | 88.69 | 33 | 18,160 |
| GSF2751-sRNA_Hfq-3 | CTCAGAAT | 5,000,014 | 100 | 100 | 88.25 | 32.9 | 427 |
| GSF2751-sRNA_Hfq-7 | GACGACAT | 3,411,220 | 100 | 100 | 88.82 | 33.02 | 1,633 |
| GSF2751-sRNA_Hfq-8 | TAATCGAT | 2,413,457 | 100 | 100 | 88.7 | 33 | 186,936 |
| GSF2751-sRNA_Hfq-9 | TCGAAGAT | 3,090,434 | 100 | 100 | 89.49 | 33.17 | 128 |

analysis (≥20 bp) having an average quality score of >32. Two of the sRNA samples from which a large number of mapped reads were obtained (>185,000) were considered for further analysis. The mapping of the reads to the *H. somni* 2336 reference genome (GenBank accession number NC_010519.1) was performed using Geneious Prime® 2021.0.3 (http://www.geneious.com). Alignment of the reads to the *H. somni* genome showed that the reads mapped to intergenic regions.

## Analysis of the intergenic regions of the *H. somni* genome

The aligned RNA-seq based transcriptome map was examined in comparison to the *H. somni* genome, and intergenic regions in the genome were identified. Promoters and terminators were predicted to confirm the identified transcripts using the online tools BPROM (Softberry Inc., Mt. Kisco, NY) and ARNold [30], respectively. A total of 100 sRNA candidates were identified at the intergenic regions (Table 3), out of which 43 have been previously reported, and of which 16 are unique to strain 2336 [8]. Of the sRNAs identified in this study, three were housekeeping sRNAs (ssrS, ffs and rnpB), which are well characterized in other bacterial strains [33–35]. The majority of the identified sRNA candidates (60%) were shorter than 200 bases (size range 53–547 bases). Promoters and rho-independent terminators were predicted for 85 and 59 sRNAs, respectively. Three of the *H. somni* strain 2336 sRNAs that were selected for further study and analyzed by Geneious Prime® are shown in Fig 1 and S1 File.

The sequencing data were screened manually using Geneious Prime®. The sequences of the sRNA candidates were comparable with those previously reported for *H. somni* 2336 [8], including those sRNAs that regulate metabolism of lysine (HS119) and glycine (HS72). The sRNAs isrK (HS144), tmRNA (HS23), and gcvB (HS166) were also identified. isrK and tmRNA encoded by the *ssrA* gene were highly expressed, as determined by RNA sequencing. Multi-sequence alignment of the *H. somni* sRNA candidate HS166 (gcvB) showed 99% identity to that of the homologous sRNA in *P. multocida* (Fig 2A). The comparative sequence analysis of HS166 to *gcvB* of *P. multocida* and *E. coli* revealed two conserved sequences (R1 and R2), which have been reported [36] (Fig 2B). Functional characterization of the above-mentioned sRNAs is needed in order to determine the exact role of these regulatory elements in *H. somni* 2336.

## Identification and characterization of sRNAs in *H. somni* 2336

The potential non-coding intergenic regions of the *H. somni* 2336 genome were identified using Geneious Prime™. Each of the sRNA candidates were manually checked for their relative read coverage level in the sequencing data and their conservation across different bacterial strains based on previous data [8]. A total of 8 sRNA candidates (HS9, HS14, HS26, HS72, HS79, HS86, HS97, HS98) were considered for further characterization based on preliminary bioinformatic analyses. All the selected sRNAs showed relatively high read coverage in the sequencing data. In addition, the sRNAs that are conserved with phylogenetically closer bacterial genomes, especially members of Pasteurellaceae family (*Mannheimia haemolytica*, *P. multocida*, *Haemophilus influenza*, etc.), but also in distantly related bacterial species, were selected with the intention of performing a comparative study. Multi-sequence analysis indicated that many of these sRNA regions in the genome may interact with the sigma-54 transcription factor (encoded by *rpoN*), which is known to control bacterial quorum sensing and virulence [38]. In the case of sigma-54 binding motifs, 11 bases are highly conserved across bacterial species [39]. For the candidates HS14, HS26, and HS98, 9 out of 11 bases of the conserved consensus bases in the sigma 54 binding motif were identical [39] (Table 4). In the case of HS9, and HS72, 8 of the 11 bases matched the conserved consensus bases in the sigma 54

**Table 3.** *Histophilus somni* strain 2336 sRNAs, their genome location, additional features, and comparative genomics.

| ID | Start#[a] | End#[a] | Length (nt) | Promoter | Rho-independent terminator | Flanking gene (left) | Flanking gene (right) | Conservation across other genome[b] | Reference |
|---|---|---|---|---|---|---|---|---|---|
| HS1 | 8109 | 8210 | 101 | -[c] | Y | HSM_0009 | HSM_0010 | C | [8] |
| HS2 | 12352 | 12414 | 63 | Y | - | HSM_0015 | HSM_0016 | ND[d] | This work |
| HS3 | 16104 | 16190 | 87 | Y | Y | HSM_0018 | HSM_0019 | B | [8] |
| HS4 | 30644 | 30884 | 240 | Y | Y | HSM_0039 | HSM_0040 | B | [8] |
| HS5 | 44663 | 45024 | 362 | Y | - | HSM_0051 | HSM_0053 | ND | This work |
| HS8 | 88943 | 89453 | 511 | Y | Y | HSM_0077 | HSM_0079 | ND | This work |
| HS9 | 91939 | 92065 | 127 | Y | Y | HSM_0081 | HSM_0082 | C | [8] |
| HS12 | 183080 | 183180 | 101 | Y | - | HSM_0171 | HSM_0172 | B | [8] |
| HS13 | 197510 | 197896 | 387 | Y | Y | HSM_0186 | HSM_0186 | B | [8] |
| HS14 | 229488 | 229780 | 293 | Y | - | HSM_0213 | HSM_0214 | C | [8] |
| HS15 | 258962 | 259117 | 156 | Y | Y | HSM_0244 | HSM_0245 | A | [8] |
| HS16 | 260385 | 260624 | 240 | Y | Y | HSM_0245 | HSM_0246 | A | [8] |
| HS18 | 261354 | 261531 | 178 | Y | Y | HSM_0246 | HSM_0247 | A | [8] |
| HS19 | 261807 | 262158 | 352 | Y | Y | HSM_0246 | HSM_0247 | A | [8] |
| HS22 | 279541 | 279732 | 192 | Y | - | HSM_0266 | HSM_0267 | B | [8] |
| HS23 | 306543 | 306993 | 451 | Y | - | HSM_0284 | HSM_0285 | D | [8] |
| HS26 | 318134 | 318443 | 310 | Y | Y | HSM_0292 | HSM_0293 | D | [8] |
| HS27 | 325426 | 325495 | 70 | Y | - | HSM_0302 | HSM_0303 | ND | This work |
| HS30 | 368822 | 369085 | 264 | Y | Y | HSM_0336 | HSM_0337 | ND | This work |
| HS31 | 382931 | 383420 | 490 | Y | - | HSM_0346 | HSM_0347 | A | [8] |
| HS32 | 385627 | 385785 | 159 | Y | Y | HSM_0348 | HSM_0349 | B | [8] |
| HS34 | 481596 | 481692 | 97 | - | Y | HSM_0415 | HSM_0416 | ND | This work |
| HS36 | 513150 | 513269 | 120 | Y | - | HSM_0443 | HSM_0014 | ND | This work |
| HS37 | 515196 | 315366 | 171 | Y | - | HSM_0016 | HSM_0017 | ND | This work |
| HS41 | 555076 | 555135 | 60 | Y | Y | HSM_0479 | HSM_0480 | ND | This work |
| HS43 | 598133 | 598401 | 269 | - | Y | HSM_0520 | HSM_0521 | ND | This work |
| HS44 | 599224 | 599432 | 209 | Y | Y | HSM_0521 | HSM_0522 | B | [8] |
| HS45 | 608001 | 608193 | 193 | Y | Y | HSM_0533 | HSM_0534 | ND | This work |
| HS46 | 616791 | 617166 | 376 | Y | - | HSM_0539 | HSM_0540 | A | [8] |
| HS47 | 617726 | 618078 | 352 | Y | - | HSM_0539 | HSM_0540 | A | [8] |
| HS48 | 618122 | 618334 | 213 | Y | Y | HSM_0539 | HSM_0540 | A | [8] |
| HS49 | 619340 | 619614 | 275 | Y | - | HSM_0540 | HSM_0541 | ND | This work |
| HS50 | 629462 | 629599 | 138 | Y | - | HSM_0550 | HSM_0019 | ND | This work |
| HS51 | 631525 | 631711 | 187 | Y | - | HSM_0021 | HSM_0022 | ND | This work |
| HS53 | 658121 | 658225 | 105 | Y | - | HSM_0565 | HSM_0566 | ND | This work |
| HS54 | 672117 | 672270 | 154 | Y | Y | HSM_0578 | HSM_0027 | ND | This work |
| HS55 | 674011 | 674147 | 137 | Y | - | HSM_0028 | HSM_0029 | ND | This work |
| HS62 | 764367 | 764500 | 134 | Y | - | HSM_0668 | HSM_0669 | ND | This work |
| HS63 | 782580 | 782672 | 93 | - | - | HSM_0688 | HSM_0689 | ND | This work |
| HS64 | 815707 | 815993 | 287 | Y | Y | HSM_0708 | HSM_0709 | ND | This work |
| HS67 | 851529 | 851687 | 158 | Y | Y | HSM_0740 | HSM_0741 | B | [8] |
| HS70 | 869792 | 869984 | 193 | Y | Y | HSM_0751 | HSM_0752 | ND | This work |
| HS72 | 876309 | 876450 | 142 | Y | Y | HSM_0658 | HSM_0659 | C | [8] |
| HS78 | 969506 | 969784 | 279 | Y | Y | HSM_0843 | HSM_0844 | ND | This work |
| HS79 | 981948 | 982242 | 295 | Y | Y | HSM_0847 | HSM_0848 | C | [8] |
| HS82 | 985484 | 985636 | 153 | Y | - | HSM_0037 | HSM_0038 | ND | This work |

(*Continued*)

**Table 3.** (*Continued*)

| ID | Start#[a] | End#[a] | Length (nt) | Promoter | Rho-independent terminator | Flanking gene (left) | Flanking gene (right) | Conservation across other genome[b] | Reference |
|---|---|---|---|---|---|---|---|---|---|
| HS83 | 987368 | 987513 | 146 | Y | - | HSM_0039 | HSM_0040 | ND | This work |
| HS86 | 1007800 | 1008014 | 215 | Y | Y | HSM_0868 | HSM_0869 | C | [8] |
| HS87 | 1008113 | 1008332 | 224 | Y | Y | HSM_0868 | HSM_0869 | A | [8] |
| HS88 | 1009543 | 1009788 | 246 | Y | Y | HSM_0869 | HSM_0870 | ND | This work |
| HS89 | 1012617 | 1012863 | 206 | Y | Y | HSM_0874 | HSM_0875 | B | [8] |
| HS90 | 1013851 | 1014063 | 213 | Y | - | HSM_0875 | HSM_0876 | ND | This work |
| HS91 | 1014280 | 1014767 | 163 | Y | Y | HSM_0875 | HSM_0876 | A | [8] |
| HS92 | 1021920 | 1022358 | 442 | Y | - | HSM_0888 | HSM_0889 | A | [8] |
| HS93 | 1022359 | 1022473 | 115 | Y | Y | HSM_0888 | HSM_0889 | A | [8] |
| HS94 | 1024358 | 1025423 | 134 | - | Y | HSM_0891 | HSM_0892 | ND | This work |
| HS96 | 1031383 | 1031557 | 175 | Y | - | HSM_0899 | HSM_0900 | ND | This work |
| HS97 | 1032012 | 1032164 | 153 | Y | Y | HSM_0900 | HSM_0901 | C | [8] |
| HS98 | 1032206 | 1032457 | 252 | Y | Y | HSM_0900 | HSM_0901 | D | [8] |
| HS105 | 1163348 | 1163469 | 122 | Y | Y | HSM_1017 | HSM_1018 | ND | This work |
| HS106 | 1180654 | 1181062 | 409 | Y | Y | HSM_1022 | HSM_1023 | ND | This work |
| HS108 | 1204957 | 1205023 | 67 | Y | Y | HSM_1047 | HSM_1048 | ND | This work |
| HS109 | 1227189 | 1227300 | 112 | Y | Y | HSM_1066 | HSM_1067 | ND | This work |
| HS101 | 1111354 | 1111410 | 57 | - | Y | HSM_0970 | HSM_0971 | ND | This work |
| HS114 | 1312678 | 1312854 | 177 | Y | Y | HSM_1143 | HSM_1144 | A | [8] |
| HS115 | 1319443 | 1319574 | 132 | Y | Y | HSM_1153 | HSM_1154 | ND | This work |
| HS117 | 1337413 | 1337658 | 246 | Y | Y | HSM_1072 | HSM_1173 | B | [8] |
| HS118 | 1368913 | 1369114 | 202 | Y | Y | HSM_1211 | HSM_1212 | ND | This work |
| HS119 | 1413711 | 1413974 | 264 | Y | - | HSM_1254 | HSM_1255 | B | [8] |
| HS120 | 1416712 | 1417102 | 392 | Y | Y | HSM_1256 | HSM_1257 | ND | This work |
| HS122 | 1464716 | 1464864 | 149 | Y | Y | HSM_1284 | HSM_1285 | ND | This work |
| HS127 | 1580074 | 1580172 | 99 | Y | Y | HSM_1383 | HSM_1384 | ND | This work |
| HS128 | 1597315 | 1597535 | 221 | Y | - | HSM_1399 | HSM_1400 | ND | This work |
| HS131 | 1659113 | 1659217 | 105 | Y | Y | HSM_1455 | HSM_1456 | ND | This work |
| HS136 | 1748564 | 1748835 | 272 | Y | Y | HSM_1521 | HSM_1522 | ND | This work |
| HS137 | 1776051 | 1776199 | 149 | Y | - | HSM_1542 | HSM_1543 | ND | This work |
| HS141 | 1806298 | 1806841 | 547 | Y | Y | HSM_1569 | HSM_1571 | ND | This work |
| HS142 | 1839525 | 1839645 | 121 | Y | Y | HSM_1590 | HSM_1591 | B | [8] |
| HS143 | 1865693 | 1865748 | 58 | Y | - | HSM_1616 | HSM_1617 | ND | This work |
| HS144 | 1874327 | 1874482 | 156 | Y | Y | HSM_1626 | HSM_1627 | ND | [8] |
| HS145 | 1875319 | 1875422 | 104 | Y | Y | HSM_1628 | HSM_1629 | ND | This work |
| HS146 | 1876066 | 1876214 | 149 | - | - | HSM_1630 | HSM_1631 | ND | This work |
| HS148 | 1922040 | 1922215 | 176 | - | - | HSM_1669 | HSM_1670 | ND | This work |
| HS150 | 1925814 | 1925932 | 118 | - | - | HSM_1675 | HSM_1676 | B | [8] |
| HS151 | 1927618 | 1928029 | 412 | Y | - | HSM_1676 | HSM_1677 | D | [8] |
| HS152 | 1928147 | 1928340 | 194 | Y | Y | HSM_1676 | HSM_1677 | A | [8] |
| HS155 | 1931457 | 1931553 | 97 | - | - | HSM_1679 | HSM_1680 | ND | This work |
| HS156 | 1938168 | 1938400 | 233 | Y | - | HSM_1688 | HSM_1689 | ND | This work |
| HS157 | 1942495 | 1942610 | 116 | Y | Y | HSM_1692 | HSM_1693 | A | [8] |
| HS159 | 1962921 | 1963057 | 137 | Y | - | HSM_1719 | HSM_1720 | ND | This work |
| HS160 | 1965414 | 1965466 | 53 | - | Y | HSM_1720 | HSM_1721 | ND | This work |
| HS161 | 1967173 | 1967232 | 60 | - | - | HSM_1722 | HSM_1723 | ND | This work |

(*Continued*)

**Table 3.** (*Continued*)

| ID | Start#[a] | End#[a] | Length (nt) | Promoter | Rho-independent terminator | Flanking gene (left) | Flanking gene (right) | Conservation across other genome[b] | Reference |
|---|---|---|---|---|---|---|---|---|---|
| HS164 | 1982114 | 1982330 | 217 | - | Y | HSM_1738 | HSM_1739 | ND | This work |
| HS166 | 2020540 | 2020731 | 192 | Y | Y | HSM_1776 | HSM_1777 | D | [8] |
| HS167 | 2025431 | 2025500 | 70 | Y | - | HSM_1782 | HSM_1783 | ND | This work |
| HS168 | 2037940 | 2038042 | 103 | Y | - | HSM_1792 | HSM_1793 | ND | This work |
| HS169 | 2071563 | 2071686 | 124 | - | - | HSM_1821 | HSM_1822 | ND | This work |
| HS172 | 2118413 | 2118518 | 106 | - | - | HSM_1867 | HSM_0060 | ND | This work |
| HS176 | 2124729 | 2124829 | 102 | Y | Y | HSM_1868 | HSM_1869 | A | [8] |
| HS179 | 2210181 | 2210317 | 137 | Y | Y | HSM_1950 | HSM_1951 | B | [8] |

[a]The start# and end# represent the boundaries of an identified transcriptionally active region (TAR), which is a potential sRNA region.

[b]sRNA sequences conserved: A—Unique to *H. somni* 2336; B–unique to *H. somni* strain 129PT; C–Phylogenetically closer to bacterial genomes of the Pasteurellaceae family (*M. haemolytica, P. multocida. H. influenzae*, etc.); D—across distant bacterial species.

[c]-Any cell with no predicted result.

[d]ND–Not determined.

binding motif. For the candidates HS86 and HS97, 7 of the 11 bases were identical to the consensus bases in the sigma 54 binding motif.

## Northern blotting to verify selected sRNA candidates

The selected 8 sRNAs candidates were chosen based on preliminary bioinformatic analysis and the relatively higher read coverage in the sequencing data. Three of these sRNA candidates (HS9, HS79, and HS97) were further selected based on bioinformatic predictions of their interaction with known virulence or biofilm genes in *H. somni*, such as *luxR* (found two *luxR* candidates in *H. somni*), *narQ, uspE*, and *uspA*. Northern blotting was used to determine the approximate size and expression level of these sRNAs, and to determine if they make any splice products. Northern blots demonstrated two bands of ~115 bases and ~60 bases for sRNA HS9, three bands of ~80 bases, ~130 bases, and ~300 bases for HS79, and two bands of ~140 bases and ~200 bases for HS97 (Fig 3). A BLAST analysis of the sequences of the sRNA-specific probes with the complete *H. somni* genome, and optimized for "somewhat similar sequences", resulted in only one hit for each of the three sRNAs (HS9, HS79, HS97). Therefore, it was unlikely that the probe bound to more than one sRNA sequence in the genome. Rather, the multiple bands for each sRNA in the Northern blots suggest multiple transcripts being synthesized from the same promoter or post-transcriptional splicing.

## Determination of the transcriptional start site of selected sRNA candidates

Two sRNA candidates, HS79 and HS97, were further characterized by 5'-RLM-RACE. The advantage of RLM-RACE over conventional RACE is that the former only amplifies the primary transcripts, and not the processed products. The bands (indicated with an oval shape) in Fig 4A were excised, cloned, and sequenced to determine the transcriptional start sites. For sRNA HS79, the transcriptional start site identified matches well with one of the bands (~300 bases) observed by Northern blotting. For HS97, the position of the transcriptional start site matched the band with a size of ~140 bases in the Northern blot. Further analysis of the promoter region of the sRNAs with the online software BPROM (Softberry Inc.) revealed additional transcriptional start sites upstream of the transcriptional start site determined by

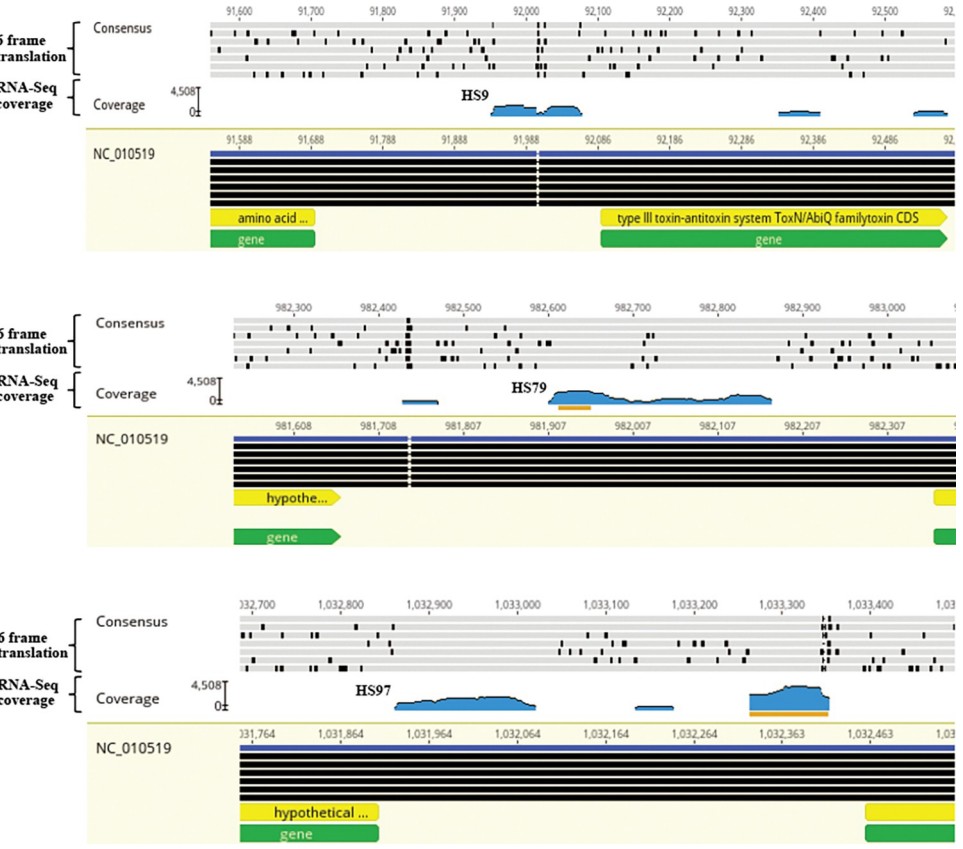

**Fig 1. Visualization of sRNAs by Geneious Prime®.** Three of the sRNAs present in the intergenic region of the genome of *Histophilus somni* strain 2336 are shown. The scale bar for the RNA sequence coverage (labelled as "coverage") depicts minimum and maximum values for the entire alignment, and the tick to its right (range 0–4508) represents the mean coverage. The consensus sequence (top-most line) is displayed above the alignment or assembly, and shows those residues that are conserved. The six-frame translation (three overlapping reading frames in the forward direction and the complementary strand in the reverse direction) of the consensus sequence is displayed just below the consensus sequence. The top, dark blue line represents the reference genome sequence, and the six black lines below it represent the six-frame translation of the reference sequence. The representative zoomed images of the six-frame translation of the reference sequence and consensus sequence is shown in S1 File. The orange lines represent the maximum coverage for the RNA sequence. The sRNA names are shown close to their corresponding sRNA transcripts in the graphs. For HS97, the first region that has the transcript mapping represents the sRNA.

RLM-RACE (Fig 4B and 4C). Schematic representation of the chromosomal regions and sequences of sRNAs HS79 and HS97 are shown in Fig 4B and 4C.

## EMSA to confirm the interaction of selected sRNAs with recombinant Hfq

To confirm the binding of sRNAs with Hfq, recombinant Hfq was expressed, purified, and incubated with *in vitro*-transcribed sRNAs HS9, HS79, and HS97 for EMSA (Fig 5). As shown, all three sRNAs specifically bound to the recombinant Hfq, and the intensity of the band shift in EMSA was proportional to the concentration of recombinant Hfq added.

## Prediction of secondary structures of selected sRNAs

Once the 5' terminus of the sRNA transcripts were assigned by RLM-RACE and examined by Northern blotting and bioinformatics, the resulting sequences were further analyzed for prediction of their secondary structures by RNAfold [31]. The predicted secondary structures of

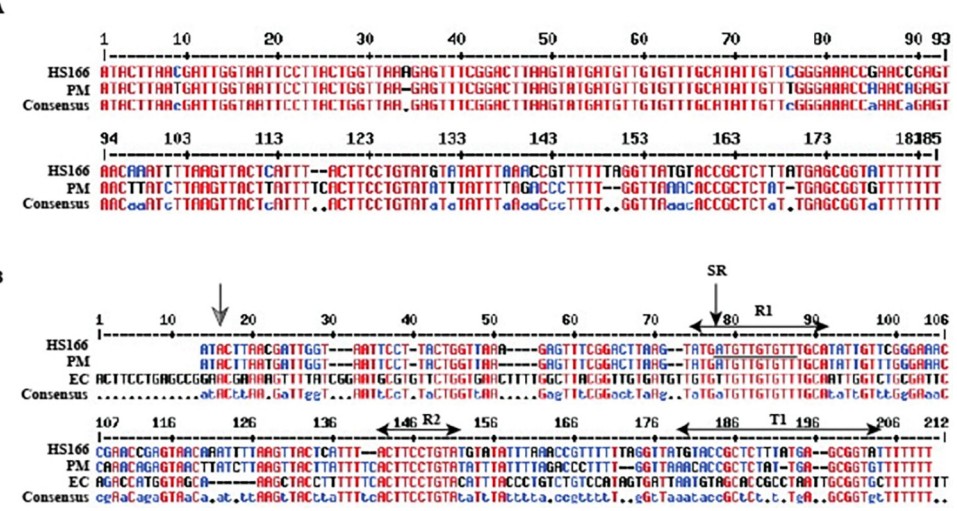

**Fig 2. Nucleotide sequence alignment of HS166 from *Histophilus somni* strain 2336 with that of other bacterial species.** (A) Sequence similarity of HS166 with *gcvB* from *Pasteurella multocida* (PM). (B) Multisequence alignment of HS166 with *gcvB* from *P. multocida* (PM) and *Escherichia coli* (EC) show sequence conservation. The conserved sequences R1 and R2 reported earlier [36] are shown above the alignment. The seed region is labelled as SR and is underlined. The predicted *H. somni gcvB* rho-independent terminator is marked as T1. The predicted start site of *P. multocida* gcvB, as determined by 5' RACE [37], is indicated by the grey arrow. All the positions in each sequence that are identical with the consensus are given in upper-case, the other positions are in lower case. The residues that are highly conserved appear in high-consensus red color. The residues that are weakly conserved are shown in low consensus blue color. Other residues are shown in neutral color, which is black. A position with no conserved residue is represented by a dot in the consensus line.

the sRNAs HS79 and HS97 are shown in Fig 6. These sRNAs contained single stranded unpaired regions rich in adenosine and uridine, and these AU-rich regions are proposed to be binding sites for Hfq in other bacteria [40].

## Discussion

Despite recent advancements in sequencing technologies for the identification of bacterial non-coding RNAs, the role of sRNAs as post-transcriptional regulators in *H. somni* is unknown. In 2012, Kumar et al. [8] identified a number of sRNAs in *H. somni* based on an RNA-Seq based transcriptome map. In that study, RNA was isolated from *H. somni* cells, enriched mRNAs, sequenced, reads were mapped to the *H. somni* genome sequence, and the

**Table 4. sRNA regions in the *H. somni* genome that can putatively interact with sigma 54.**

| ID | Sigma 54 binding motif* |
|----|------------------------|
| HS9 | TGCCACAAGTGCTACT |
| HS14 | TAGCCCATTGCTTGCA |
| HS26 | TAGCACTTGCCGATTGCC |
| HS72 | TGGCGAGGACTGTTACT |
| HS86 | AGCCACTTTCGACAGTGGCG |
| HS97 | TCCCCCTCTCGTTGAA |
| HS98 | CGGCACCCTATTGAA |

*Conserved sequences matching with established sigma 54 binding motif (5'-TGGCACG-N4-TTGCW-3') are underlined.

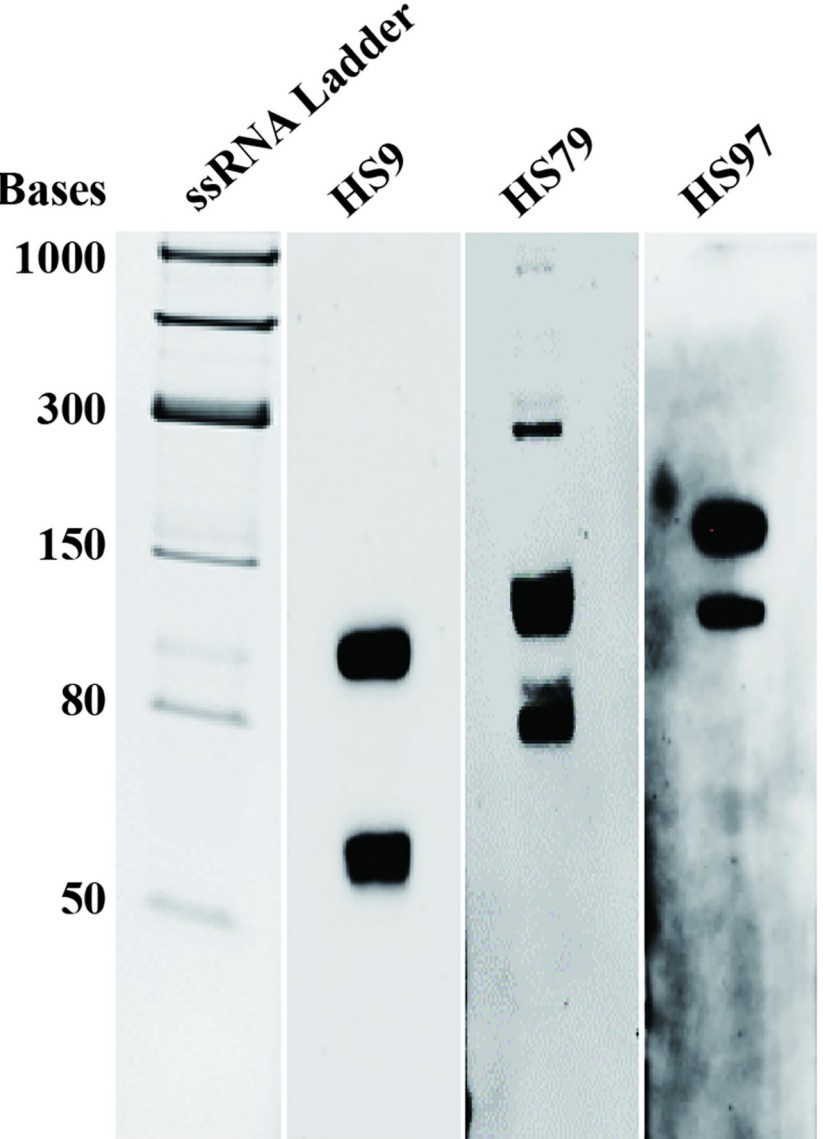

**Fig 3. Northern blot showing the approximate size of sRNAs HS9, HS79, and HS97.** Two bands of ~115 bases and ~60 bases were identified for sRNA HS9, three bands of ~80 bases, ~130 bases and ~300 bases for HS79, and two bands of ~140 bases and ~200 bases for HS97. The multiple bands in the Northern blot are likely indicative of post-transcriptional processing events or multiple transcripts synthesized from the same promoter.

intergenic regions were analyzed to identify potential sRNAs. A total of 94 sRNAs were identified in the *H. somni* genome, out of which some sRNAs were unique to *H. somni* strain 2336 compared to the non-virulent strain 129 Pt [8]. However, none of the predicted sRNAs in *H. somni* were validated or further characterized in the above study. In our current study, the sRNAs that bind to the global regulatory protein Hfq were isolated by co-immunoprecipitation with anti-Hfq antibody, followed by sequencing, and further bioinformatic and laboratory analysis to identify and partially characterize potential sRNA candidates in *H. somni*. Hfq is known to contribute to regulation of important bacterial traits that include quorum sensing, virulence, pathogenesis, biofilm formation etc. in many Gram-negative bacteria [41–43]. Thus, the identification and characterization of sRNAs associated with Hfq will be valuable in

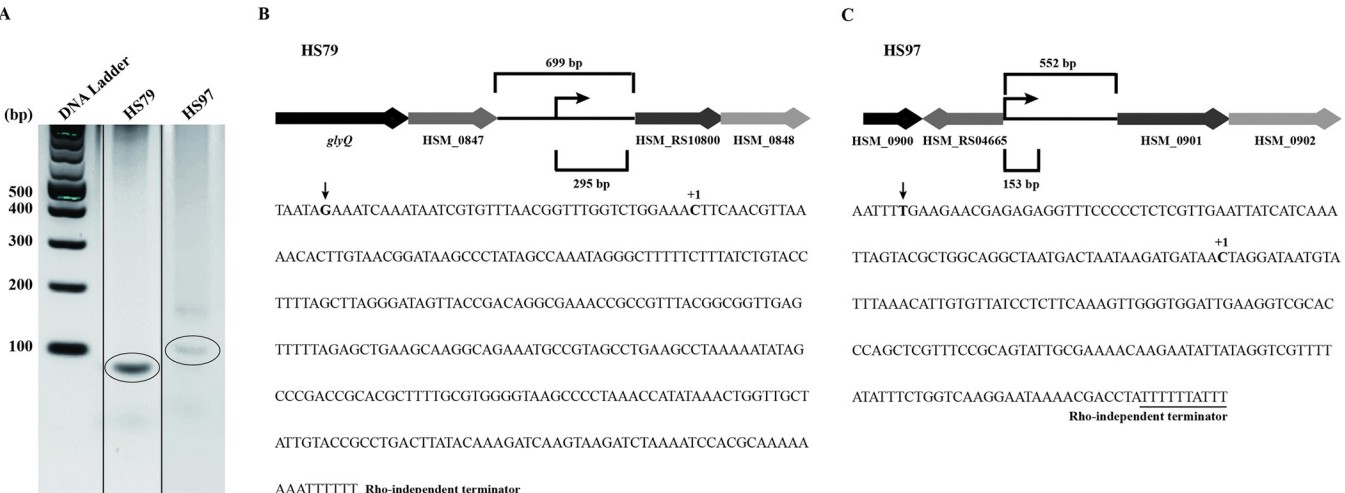

**Fig 4. Identification of the transcriptional start sites and schematic representation of the chromosomal regions of sRNAs.** (A) Identification of the transcriptional start sites of sRNAs HS79 and HS97 by RLM-RACE. The DNA bands considered for cloning and sequencing are circled. (B) Schematic representation of the chromosomal regions of sRNAs HS79, and (C) HS97. The genes flanking the sRNA sequence region are also shown. The nucleotide sequence for the sRNAs HS79 (295 bases) and HS97 (153 bases) are given below the schematic representation. The transcriptional start site identified by RLM-RACE is indicated by +1. An additional, upstream transcriptional start site was also predicted and is indicated by the arrow. The predicted rho-independent terminator is underlined.

regard to understanding post-transcriptional regulation of genes involved in virulence and pathogenesis in *H. somni*. Out of the 100 intergenic sRNA regions identified in this study, 43 have been previously reported [8]. Kumar et al. identified a number of sRNAs that were not found in our study [8], but other sRNAs were identified in our study that were not detected by Kumar et al. [8]. The differences between the sRNAs identified in our study with that of Kumar et al. [8] could be due to the difference in culture conditions, the RNA extraction method, the selected interaction with Hfq, or the RNA type used for sequencing.

Northern blotting was carried out to determine the approximate size, abundance, and splice products of any of the selected sRNAs. Multiple bands in Northern blots of sRNAs is an indication of more than one transcript being synthesized from the same promoter or post-transcriptional splicing. Hfq is structurally and sequence-wise closely related to the Sm/Lsm family of

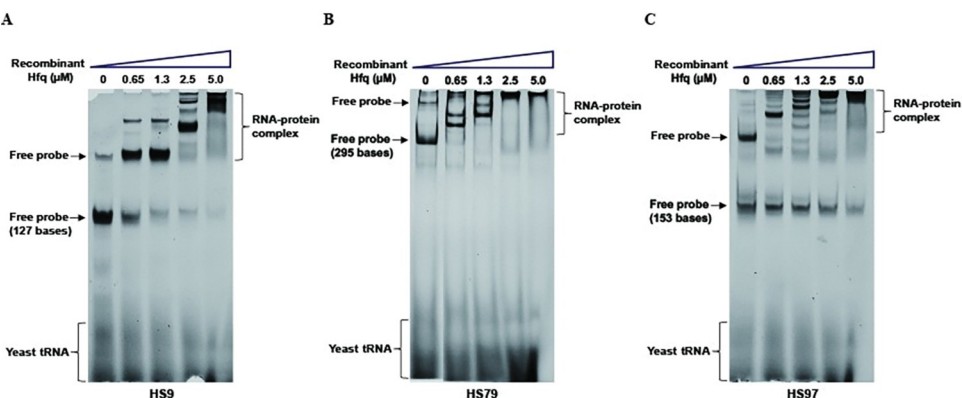

**Fig 5. EMSA confirmation of the binding of Hfq with sRNAs.** sRNAs HS9 (A), HS79 (B), and HS97 (C) were transcribed *in vitro* and treated with varying concentrations of recombinant Hfq, followed by native polyacrylamide gel electrophoresis and staining. The shifts in the sRNA bands are due to interaction with Hfq. The upper band in the gels is free probe and is due to the secondary structure formation.

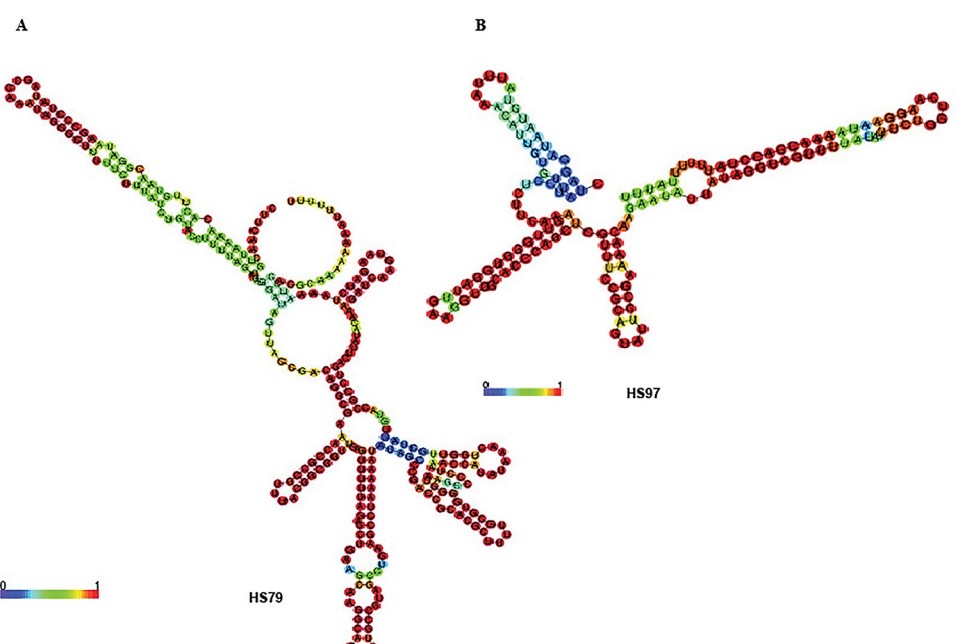

**Fig 6. Predicted secondary structure of sRNAs.** The secondary structure of sRNAs HS79 (A) and H97 (B) was predicted by RNAfold. The optimal secondary structure with a minimum free energy (MFE) is shown. Base pairing probability is indicated by color palette ranging from 0 to 1 (purple to red).

proteins involved in splicing in eukaryotes. Thus, it has been suggested that Hfq may be functionally similar to eukaryotic Sm proteins [44]. EMSA experiments with selected sRNA candidates confirmed that these sRNAs specifically bound to Hfq, supporting that Hfq likely contributes a supporting role for these sRNAs in *H. somni*. Transcriptional start sites were determined for selected sRNAs, which further enabled the structural prediction of these sRNAs. The secondary structure (containing AU-rich regions and several stem loops) of the selected sRNAs supported that these sRNAs interacted with Hfq, and the potential role of these sRNAs in the control of various biological processes [40, 45]. Information on rho-independent terminators could not be obtained for many of the sRNAs identified using any of the online prediction tools, and further analysis is needed to categorize them as either novel sRNAs or novel genes.

A number of sRNAs, including isrK, tmRNA, gcvB and those that regulate metabolism of lysine and glycine were identified in this study and these results correlate with previous results on *H. somni* sRNAs [8]. Most of these sRNAs are known to play important regulatory roles in virulence, pathogenesis, acid resistance, thermotolerance, and osmotic stress response in many Gram-negative bacteria [46–51]. In *S. enterica* serovar *typhimurium*, isrK is expressed in the stationary phase under conditions known to regulate invasion, including low pH, low magnesium levels, and low oxygen [46]. tmRNA contributes to virulence and pathogenesis in many bacterial pathogens including *Legionella pneumophilia*, *S. enterica* serovar *typhimurium*, *Francisella tularensis*, *P. aeruginosa*, and others [47–50]. GcvB is known to regulate a number of amino acid transport systems, as well as amino acid synthesis genes in a number of pathogenic bacteria, including *Yersinia pestis*, *Haemophilus influenzae*, *V. cholerae*, *S. enterica* serovar *typhimurium*, *Klebsiella pneumoniae*, and *P. multocida* [37, 52–56]. GcvB contributes to important regulatory roles in oxidative stress, biofilm formation, and lipid A and lipopolysaccharide (LPS) modifications in Gram-negative bacteria [57–60]. In *P. multocida*, another

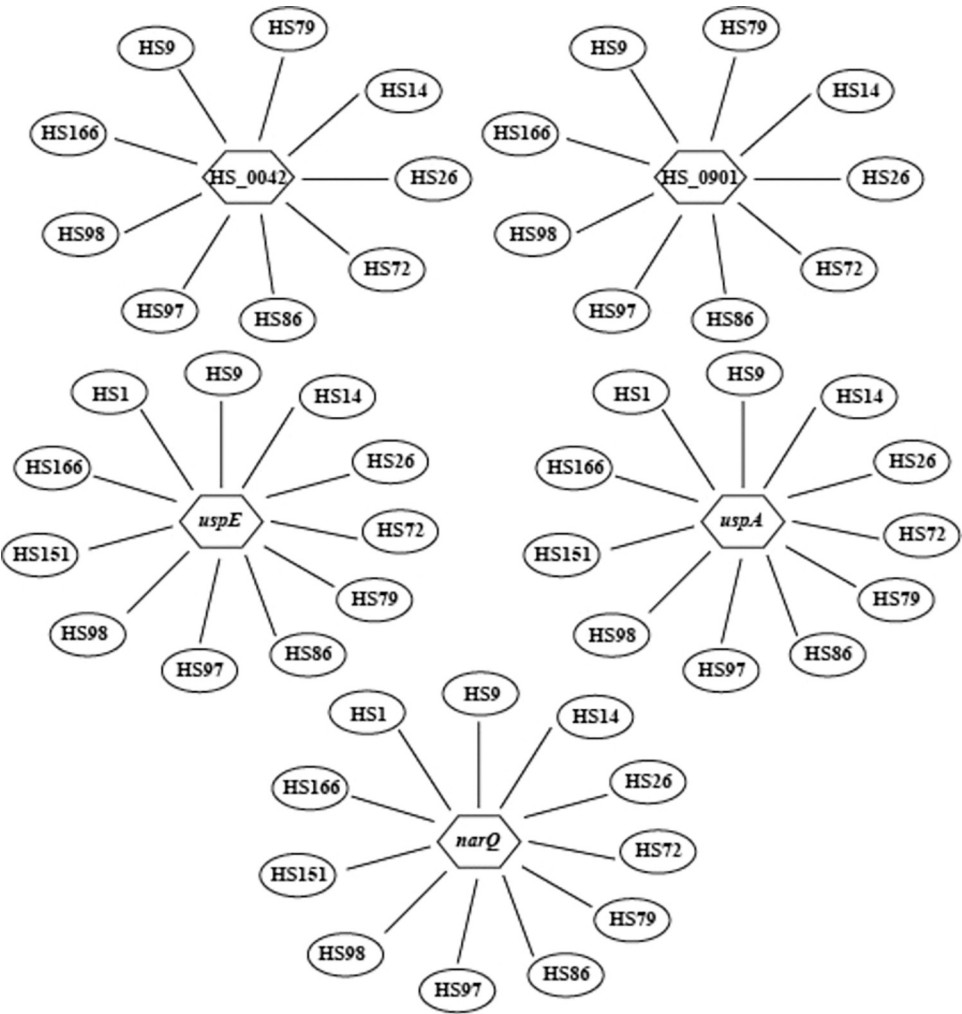

**Fig 7. Predicted targets of candidate sRNAs.** sRNA candidates that could interact with genes putatively involved in *H. somni* biofilm formation and virulence (*uspE*, *uspA*, *luxR* [HS_0901], *narQ*, *luxR* [HS_0042]). *uspE* encodes a stress response regulator and is required for virulence; *uspA* encodes a nonspecified stress protein; *luxR* (HS_0901) encodes a regulatory protein, sigma-70 region 4 type 2; *narQ* encodes a nitrite/nitrate sensor kinase; *luxR* (HS_0042) encodes a nitrate/nitrite response regulator.

bovine respiratory pathogen that is closely related to *H. somni*, the overexpression of gcvB results in increased lag phase growth [37]. Further characterization of gcvB in *H. somni* is needed to understand the regulatory role of gcvB in the virulence of this respiratory pathogen.

Target predictions were made for the 8 sRNAs identified with high read coverage in the sequencing data using IntaRNA [45]. This analysis revealed a number of interesting gene candidates, including the LuxR family candidate HS_0042 (nitrate/nitrite response regulator), *narQ* (encoding a nitrite/nitrate sensor kinase), LuxR family candidate HS_0901 (regulatory protein, sigma-70 region 4 type 2), *uspE* (encoding a stress response regulator), and *uspA* (encoding a stress protein) (Fig 7). The majority of these target genes bound to the typical A/U rich regions found in small RNAs. sRNAs are known to interact with multiple target genes, and their interaction is through the internal A/U rich regions and the 3' U-rich region after the transcriptional terminator [61]. In addition, some sRNAs were predicted to interact with their target genes at the UG-rich regions and the role of UG-rich regions in binding to target

mRNAs has been reported [62]. The stress protein UspA plays a crucial role in the regulation of stress resistance and virulence in *Salmonella* spp. [63]. Our group has reported that UspE is a global regulator controlling virulence and biofilm formation in *H. somni* [64]. The LuxR family candidate HS_0901 (regulatory protein, sigma-70 region 4 type 2) and LuxR family candidate HS_0042 (nitrate/nitrite response regulator) are yet to be characterized in *H. somni*. Recently, Martín-Rodríguez *et al*. reported that NarQ, along with the nitrate response regulator NarP, plays a regulatory role in biofilm formation by uropathogenic *E. coli* [65]. Thus, the functional characterization of the nitrate response regulator HS_0042 (LuxR family candidate) may identify its role in the regulatory network of biofilm formation in *H. somni*. In addition, further studies on the sRNAs that are predicted to be associated with these important regulatory genes would facilitate future studies to develop new promising strategies to effectively prevent diseases due to *H. somni*.

## Conclusions

In this study, a total of 100 *H. somni* strain 2336 putative sRNAs, including previously unknown sRNAs, were identified that are associated with the global regulatory protein Hfq under the growth conditions tested. Highly conserved noncoding sRNAs, including the housekeeping sRNAs ssrS, ffs, and rnpB, and tmRNA were also identified. Bioinformatic analysis revealed that many of the sRNA candidates could potentially interact with genes involved in quorum sensing, virulence, and biofilm formation in many bacterial pathogens. This is the first study in which Hfq-associated sRNAs were identified and characterized in *H. somni*. Further functional characterization of these Hfq-associated sRNAs may shed light into the posttranscriptional regulatory network, and the role of Hfq and sRNAs in the virulence and pathogenicity of *H. somni*.

## Supporting information

**S1 File. This file contains all the supporting text and figure.**
(DOCX)

**S1 Raw images.**
(PDF)

## Acknowledgments

We thank Christopher McAllister for assistance with inoculation, bleeding, and sedation of the rats.

## Author Contributions

**Conceptualization:** Thomas J. Inzana.

**Data curation:** Bindu Subhadra, Dianjun Cao, Roderick Jensen.

**Formal analysis:** Bindu Subhadra, Roderick Jensen, Clayton Caswell, Thomas J. Inzana.

**Funding acquisition:** Clayton Caswell, Thomas J. Inzana.

**Investigation:** Bindu Subhadra, Dianjun Cao, Thomas J. Inzana.

**Methodology:** Bindu Subhadra, Dianjun Cao, Roderick Jensen, Clayton Caswell, Thomas J. Inzana.

**Project administration:** Thomas J. Inzana.

**Resources:** Thomas J. Inzana.

**Supervision:** Dianjun Cao, Thomas J. Inzana.

**Validation:** Bindu Subhadra.

**Writing – original draft:** Bindu Subhadra.

**Writing – review & editing:** Dianjun Cao, Roderick Jensen, Clayton Caswell, Thomas J. Inzana.

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
