## [Decision Letter · Decision Letter 0]

9 Jan 2023

PONE-D-22-34191

Identification and initial characterization of Hfq-associated sRNAs in Histophilus somni strain 2336v

PLOS ONE

Dear Dr. Inzana,

Thank you for submitting your manuscript to PLOS ONE. After careful consideration, we feel that it has merit but does not fully meet PLOS ONE’s publication criteria as it currently stands. Therefore, we invite you to submit a revised version of the manuscript that addresses the points raised during the review process.

We look forward to receiving your revised manuscript.

Kind regards,

Marty Roop

Academic Editor

PLOS ONE

Journal Requirements:

"This work was supported by USDA-NIFA grant 2017-67015-26797 to TJI, and funds from Long Island University."

Additional Editor Comments:

Two external reviewers have evaluated your paper and their comments are provided below. As you can see, both are positive about the work, but both also express the opinion that it is 'incomplete' in its current state. Consequently, I am going to ask that you submit a revised version of the paper that adequately and appropriately addresses all of the comments raised by both reviewers.

Reviewers' comments:

Reviewer's Responses to Questions

Comments to the Author

1. Is the manuscript technically sound, and do the data support the conclusions?

Reviewer #1: Yes

Reviewer #2: Yes

2. Has the statistical analysis been performed appropriately and rigorously?

Reviewer #1: Yes

Reviewer #2: Yes

3. Have the authors made all data underlying the findings in their manuscript fully available?

Reviewer #1: Yes

Reviewer #2: Yes

4. Is the manuscript presented in an intelligible fashion and written in standard English?

Reviewer #1: Yes

Reviewer #2: Yes

5. Review Comments to the Author

Reviewer #1: This is a well performed study that describes a set of sRNA molecules in Histophilus somni that were identified as interacting with the RNA chaperone Hfq.

While there are no major comments about the experimental design or data, this reviewer feels that the manuscript leaves the reviewer hungry, raising many unanswered questions. Solving this would need a considerable amount of additional experimental work. It is suggested that the authors make the manuscript more concise and direct by shortening the methods and that some of the speculative parts of the results and discussion are also reduced.

Minor comments, corrections and suggestions for shortening the text

1 Is Fig 1 really needed? Also, the title is not correct, this is ‘detection’, not ‘identification’.

2 IntaRNA does not indicate or show that a sRNA can bind to a gene, it suggests or predicts this. Please change this in the text, or show experimental evidence.

3 There is an error in Fig5, why is the panel with uspE in the centre shown twice

4 Are there any sequence motifs that are common to the multiple targets that could explain the promiscuous targeting of these genes numerous RNAs?

5 For section 3.2. (Transcriptome-wide analyses of RNA sequencing results); ‘transcription wide’ is not appropriate, this is mapping of identified RNA sequences to the Hs genome.

6 Fig 4 shows that some of the suggests that there are either alternative start sites for these molecules, or that they are processed. Tha authors write that RLM-RACE ‘only amplifies the primary transcripts, and not the processed products’. For HS97, why is the longer (so unprocessed?) not identified by this technique (L104, Fig 6). Do the authors have RNAseq data from this strain, which will allow them to determine whether the different transcripts are found? If not, they should mine the data from ref 8.

7 What do the RNA secondary structures bring to the manuscript. Here, experimental work is needed to show that this is of importance.

8 The methods can be shortened; many of the procedures are standard or have been described in detail in previous work by the authors, including the overexpression and purification of Hfq, producing the antibodies, pulldown and purification of Hfq associated RNA, in vitro transcription and northern blotting(here I assume that you mean a 15% acrylamide gel and not a 15% Urea as indicated in the text!).

9 The half page of conclusions is redundant.

Reviewer #2: In the manuscript “Identification and initial characterization of Hfq-associated sRNAs in Histophilus somni strain 2336” the authors outline a pull down procedure whereby they identify sRNAs that interact with the chaperone protein Hfq in the bacteria Histophilus somni. They go on to perform some initial characterization experiments on a select number of the identified sRNAs, including northern blots, mapping of transcription start sites, and confirmation of interactions with Hfq by EMSA.

Overall the work outlined is good, experiments are well controlled, performed appropriately and the manuscript is well written.

That being said, the study feels incomplete. The manuscript contains several sections that are based on computer predictions rather than experimental data and the amount of new experimentally generated data in the manuscript is relatively low. Follow up characterization is only performed for 2 or 3 sRNAs and no experiments are performed to investigate the role or function of any of them. The study would be enhanced if kore follow up work was performed on a larger number of the novel identified sRNAs.

In addition to the comments above the following issues need to be addressed.

Line 31: Please remove the sentence “to initiate understanding their role in regulation of virulence factors” as the role of these sRNAs in regulating virulence factors is speculative.

Line 36: replace the word “depicted” with suggests

Line 64:… a class of regulatory RNAs ranging in size…” please insert “typically” in this sentence as sRNAs can be outside this size range

Line 70 “…which plays crucial roles in a variety…”

Line 101 The LB in LB medium stands for lysogeny broth not luria-bertani

Line 235: Please provide detailed information regarding the composition, size, and nature of the biotinylated probes used for northern blotting.

Line 270: Six independent sRNA preparations were used for deep sequencing yet on line 281 it states that two of the sRNA samples from which a large number of reads were obtained were considered for further analysis. Can the authors explain exactly how many samples were sequenced and how those datasets were processed? This is unclear.

Line 291: the authors should elaborate on the process that led to the identification of 100 sRNA candidates. What criteria were used, what cut offs were applied when analyzing the promoter and terminator predictions? This information could be included as a supplemental to the materials and methods.

Line 300: The figure legend for figure 2 needs to be greatly expanded. For example, please explain the significance of “six frame translation” and also the significance of the six black bars and one dark blue bar running from left to right across each figure.

Line 332: Please add more detail to the figure legend for figure 3. Indicate the significance of the color coding of residues. What is the significance of uppercase versus lowercase letters? The rho independent terminator T1 should be more clearly labeled. Specifically which residues are encompassed by T1.

Line 345: “A total of 8 sRNA candidates (HS9, HS14, HS26, HS72, HS79, HS86, HS97, HS98) were considered for further characterization based on preliminary bioinformatic analyses”. A detailed account of the selection criteria used to identify these 8 candidate sRNAs should be included. Why were these 8 selected. This information could be included as supplemental materials.

Line 356: Please add the Sigma 54 consensus binding motif sequence to Table 3

Line 360: ” Based on the interaction with known genes that affect virulence and biofilm formation in Gram-negative bacteria…” What interactions and what genes are the authors referring to? The reasons for selecting these three sRNAs for further study are not clear. Furthermore, while 3 sRNAs were chosen for further analysis (HS9, HS79, and HS97) in some experiments data is only provided for two of them (HS79, and HS97 in Fig 6 and Fig 8) while in others there is data for all three (Fig 4, and Fig 7). In addition to explaining why these three were selected for further analysis please explain this inconsistency and/or provide data for HS9 in Fig 6 and Fig 8.

Line 369: “…in the Northern blots indicated multiple transcripts…” replace “indicated” with “suggests”

Line 363, Fig 4, and throughout the manuscript: When referring to the size of ssRNA molecules it is more accurate to refer to size in “bases” or “nucleotides (nt)” rather than “base pairs (bp)”.

Line 378: Target predictions using IntaRNA (or other target prediction software) are notoriously unreliable. Unless accompanied by experimental validation, these analyses are merely suggestive. Furthermore the details regarding how the IntaRNA analysis was performed are unclear. Consequently, I suggest removing section 3.6 from the results section. Some of the more interesting predicted interactions could be mentioned in the discussion section.

Line 401: Why were only 2 sRNAs investigated?

Line 410: What is the significance of the nucleotide residues in bold highlighted with an arrow in figure 6B and figure 6C?

Line 419: In the EMSA experiments shown in Fig 7, why are two bands of differing sizes detected for the in vitro transcribed RNA probes in all three experiments? Were the in vitro transcribed RNAs purified and DNAse treated? How was removal of DNA confirmed? Also it would be useful to include labels indicating the size and identity of each RNA molecule in Fig 7

Line 431: “Once the 5’ and 3’ termini of the sRNA transcripts were assigned by RLM-RACE…” Identification of 3’ ends by RACE was not carried out. Please clarify this statement and/or add the 3’ RACE data.

Fig 8: The figure provided is extremely low resolution and blurry. Please provide high quality figures. Also the two individual RNAs should be labeled in the diagram.

Line 436: “The complex RNA-fold structures containing several stem loops in conjunction with the multiple target genes predicted by IntaRNA [40] is indicative of the potential of these sRNAs to control diverse biological processes.” This statement is a bit of an overreach and should be removed.

Line 474: “The secondary structure (containing AU-rich regions and several stem loops) of the selected sRNAs confirms…” Please change the word “confirms” as the predicted secondary structures by themselves do not conclusively demonstrate anything.

Line 484 capitalize the G in gram negative. Same thing on line 494

Line 485 “including pH”? Which pH are they referring to?

Lines 480 to 505: This entire section of the discussion seems superfluous. It does not discuss any of the results generated in this manuscript, rather it discusses the roles of some small RNA's in other bacteria. It is unclear why this is included or if it is necessary.

The authors should provide a more in-depth discussion of the similarities and differences between the sRNAs identified in this study and the study of Kumar et al (reference 8). For example, were the growth conditions similar? Can they speculate why certain sRNAs were identified in one study over another.

6. PLOS authors have the option to publish the peer review history of their article (what does this mean?). If published, this will include your full peer review and any attached files.

Do you want your identity to be public for this peer review? For information about this choice, including consent withdrawal, please see our Privacy Policy.

Reviewer #1: No

Reviewer #2: No

---

## [Author Response · Author response to Decision Letter 0]

23 Feb 2023

Reviewer #1: This is a well performed study that describes a set of sRNA molecules in Histophilus somni that were identified as interacting with the RNA chaperone Hfq.

While there are no major comments about the experimental design or data, this reviewer feels that the manuscript leaves the reviewer hungry, raising many unanswered questions. Solving this would need a considerable amount of additional experimental work. It is suggested that the authors make the manuscript more concise and direct by shortening the methods and that some of the speculative parts of the results and discussion are also reduced.

Thank you for each of your valuable suggestions. We have shortened the methods, results and discussion sections to make the manuscript more concise.

Minor comments, corrections and suggestions for shortening the text

1 Is Fig 1 really needed? Also, the title is not correct, this is ‘detection’, not ‘identification’.

We have removed this figure and changed the title to begin with “Detection”.

2 IntaRNA does not indicate or show that a sRNA can bind to a gene, it suggests or predicts this. Please change this in the text, or show experimental evidence.

We have taken out this paragraph.

3 There is an error in Fig5, why is the panel with uspE in the centre shown twice

We have corrected now Fig. 7 (new figure no.).

4 Are there any sequence motifs that are common to the multiple targets that could explain the promiscuous targeting of these genes numerous RNAs?

The majority of the target genes were binding to the typical A/U-rich regions found in small RNAs. It has been reported that the sRNAs can interact with multiple target genes, and their interaction is through the internal A/U rich regions and the 3’ U-rich region after the transcriptional terminator (Andrade JM et al. 2013, PlusOne). In addition, some sRNAs interacted with their target genes at the UG-rich regions and the role of UG-rich regions in binding to target mRNAs has been reported earlier (Lukavsky PJ et al. 2013, Nat. Struct. Mol. Biol.). I have added this information with the references in the revised manuscript.

5 For section 3.2. (Transcriptome-wide analyses of RNA sequencing results); ‘transcription wide’ is not appropriate, this is mapping of identified RNA sequences to the Hs genome.

We have corrected the statement in the text.

6 Fig 4 shows that some of the suggests that there are either alternative start sites for these molecules, or that they are processed. The authors write that RLM-RACE ‘only amplifies the primary transcripts, and not the processed products’. For HS97, why is the longer (so unprocessed?) not identified by this technique (L104, Fig 6). Do the authors have RNAseq data from this strain, which will allow them to determine whether the different transcripts are found? If not, they should mine the data from ref 8.

We referred to the paper suggested by the reviewer; however, we couldn’t find any information on the longer different transcript at the concerned location. However, an additional transcriptional start site upstream of the start site identified by RLM-RACE was predicted by the online tool, which is indicated by ‘arrow’ in Fig. 4C (new figure no.). We have added information about the additional transcriptional start site in the text, and also have modified the legend for Fig. 4 (new figure no.).

7 What do the RNA secondary structures bring to the manuscript? Here, experimental work is needed to show that this is of importance.

We included the predicted secondary structure of selected sRNAs as part of the preliminary analysis. However, the AU-rich regions in the secondary structure of sRNAs are also indicative of the binding sites for Hfq and other target genes, and therefore important.

8 The methods can be shortened; many of the procedures are standard or have been described in detail in previous work by the authors, including the overexpression and purification of Hfq, producing the antibodies, pulldown and purification of Hfq associated RNA, in vitro transcription and northern blotting (here I assume that you mean a 15% acrylamide gel and not a 15% Urea as indicated in the text!).

We have modified the methods session. We have deleted the detailed description of the standard protocols and given proper reference. Also, the previous paper by us detailing all the protocols was a pre-print version by Research Square.

Yes, 15% acrylamide gel. We have now specified this in the text.

9 The half page of conclusions is redundant.

We have made the ‘conclusion’ more concise.

Reviewer #2: In the manuscript “Identification and initial characterization of Hfq-associated sRNAs in Histophilus somni strain 2336” the authors outline a pull down procedure whereby they identify sRNAs that interact with the chaperone protein Hfq in the bacteria Histophilus somni. They go on to perform some initial characterization experiments on a select number of the identified sRNAs, including northern blots, mapping of transcription start sites, and confirmation of interactions with Hfq by EMSA.

Overall the work outlined is good, experiments are well controlled, performed appropriately and the manuscript is well written.

That being said, the study feels incomplete. The manuscript contains several sections that are based on computer predictions rather than experimental data and the amount of new experimentally generated data in the manuscript is relatively low. Follow up characterization is only performed for 2 or 3 sRNAs and no experiments are performed to investigate the role or function of any of them. The study would be enhanced if kore follow up work was performed on a larger number of the novel identified sRNAs.

Thank you for all of your valuable comments. We tried to characterize selected sRNAs to understand their role in various phenotypic and genotypic characteristics of Histophilus somni. We attempted to over-express the sRNAs using different vector systems suitable for this pathogen. However, over-expression of the sRNAs in H. somni was not successful, as this is a difficult pathogen to work with genetically due to a very tight restriction-modification system. In addition, the over-expression tools available for this organism are very limited. We are working on knocking out one or more of the more characterized sRNAs that are predicted to affect genes required for virulence. However, that will be a major manuscript in itself, and this work is an important step toward that goal.

In addition to the comments above the following issues need to be addressed.

Line 31: Please remove the sentence “to initiate understanding their role in regulation of virulence factors” as the role of these sRNAs in regulating virulence factors is speculative.

We have modified the text.

Line 36: replace the word “depicted” with suggests

We replaced the word ‘depicted’.

Line 64:… a class of regulatory RNAs ranging in size…” please insert “typically” in this sentence as sRNAs can be outside this size range

We have inserted ‘typically’ in the sentence.

Line 70 “…which plays crucial roles in a variety…”

We have modified it.

Line 101 The LB in LB medium stands for lysogeny broth not luria-bertani

We have corrected it.

Line 235: Please provide detailed information regarding the composition, size, and nature of the biotinylated probes used for northern blotting.

Sample Barcode sequence PF Clusters % Perfect % PF % >= Q30 Mean Quality

 barcode Clusters bases Score Mapped Reads

GSF2751-sRNA_Hfq-1 CTATACAT 4,416,420 100 100 88.01 32.85 633,747

GSF2751-sRNA_Hfq-11 TCGGCAAT 2,667,639 100 100 88.69 33 18,160

GSF2751-sRNA_Hfq-3 CTCAGAAT 5,000,014 100 100 88.25 32.9 427

GSF2751-sRNA_Hfq-7 GACGACAT 3,411,220 100 100 88.82 33.02 1,633

GSF2751-sRNA_Hfq-8 TAATCGAT 2,413,457 100 100 88.7 33 186,936

GSF2751-sRNA_Hfq-9 TCGAAGAT 3,090,434 100 100 89.49 33.17 128

The sequence of the biotinylated probes were given in Table 1. We have modified the text under ‘Materials and Methods – Northern blotting’.

Line 270: Six independent sRNA preparations were used for deep sequencing yet on line 281 it states that two of the sRNA samples from which a large number of reads were obtained were considered for further analysis. Can the authors explain exactly how many samples were sequenced and how those datasets were processed? This is unclear.

A total of six sRNA preparations were used for deep sequencing. Out of the six sRNA samples, two samples (GSF2751-sRNA_Hfq-1 & GSF2751-sRNA_Hfq-8) having the highest number of mapped reads aligned to H. somni genome were considered for further analysis. The mapping of the reads to the H. somni genome was performed by Geneious Prime. We have added this information in the text.

The below table shows the RNA sequencing result summary.

Line 291: the authors should elaborate on the process that led to the identification of 100 sRNA candidates. What criteria were used, what cut offs were applied when analyzing the promoter and terminator predictions? This information could be included as a supplemental to the materials and methods.

The mapping of the sRNA reads to H. somni 2336 reference genome was performed by Geneious Prime. The aligned RNA-sequence based transcriptome map was manually screened to identify intergenic regions. The intergenic regions were also compared with those previously reported for H. somni 2336 [reference 8]. Promoters and terminators were predicted to confirm the identified transcripts (intergenic regions) using online tools BPROM (Softberry Inc., Mt. Kisco, NY) and ARNold [reference 30], respectively. To increase the specificity, BPROM was run on the sequence upstream of the identified transcripts within the region between two ORFs. BPROM was run with the default settings. ARNold finds rho-independent terminators in nucleic acid sequences. The search procedure used two complementary programs, Erpin and RNAmotif. The intergenic region between two ORFs was considered for identifying the rho-independent terminator and the default settings for the prediction tool was applied. The above information is included as a supplemental to the ‘materials and methods’ (labelled as S1 Supporting Information) as suggested.

Line 300: The figure legend for figure 2 needs to be greatly expanded. For example, please explain the significance of “six frame translation” and also the significance of the six black bars and one dark blue bar running from left to right across each figure.

The consensus sequence is displayed above the alignment or assembly, and shows which residues are conserved, and which residues are variable. The 6 frame translation of the consensus sequence is given just below it. The dark blue line indicates the reference sequence gene, and the six black lines below the blue line indicates the 6 frame translation of the reference sequence. The Fig 1 (new figure no.) legend has been elaborated.

Line 332: Please add more detail to the figure legend for figure 3. Indicate the significance of the color coding of residues. What is the significance of uppercase versus lowercase letters? The rho independent terminator T1 should be more clearly labeled. Specifically which residues are encompassed by T1.

All the positions in each sequence that are identical with the consensus are in upper-case, the other positions are in lower case. The residue that is highly conserved appears in high-consensus color which is red. A residue that is weakly conserved appears in low consensus color and is shown in blue. Other residues appear in neutral color which is black. A position with no conserved residue is represented by a dot in the consensus line. We have modified the figure legend and the T1 has been clearly marked in Fig. 2 (new figure no.).

Line 345: “A total of 8 sRNA candidates (HS9, HS14, HS26, HS72, HS79, HS86, HS97, HS98) were considered for further characterization based on preliminary bioinformatic analyses”. A detailed account of the selection criteria used to identify these 8 candidate sRNAs should be included. Why were these 8 selected. This information could be included as supplemental materials.

We selected a number of sRNAs which showed relatively high expression level in the sequencing data for further studies. Also, our initial plan was to do a comparative study by selecting a number of sRNAs conserved in phylogenetically closer bacterial genomes especially members of Pasteurellaceae family (Mannheimia haemolytica, P. multocida, Haemophilus influenzae, etc.) and also in distantly related bacterial species. Thus, the 8 sRNAs selected for further studies were belonging to those conserved in phylogenetically closer genomes or distantly related genomes. We have updated this information in the text.

Line 356: Please add the Sigma 54 consensus binding motif sequence to Table 3.

We have added the sigma 54 consensus binding motif 5'-TGGCACG-N4-TTGCW-3' to the Table 3 suffix.

Line 360: ” Based on the interaction with known genes that affect virulence and biofilm formation in Gram-negative bacteria…” What interactions and what genes are the authors referring to? The reasons for selecting these three sRNAs for further study are not clear. Furthermore, while 3 sRNAs were chosen for further analysis (HS9, HS79, and HS97) in some experiments data is only provided for two of them (HS79, and HS97 in Fig 6 and Fig 8) while in others there is data for all three (Fig 4, and Fig 7). In addition to explaining why these three were selected for further analysis please explain this inconsistency and/or provide data for HS9 in Fig 6 and Fig 8.

We were interested in identifying some sRNAs that are important for virulence and biofilm formation in H. somni for further studies. We checked the interaction of 8 sRNA candidates selected initially with the known biofilm/virulence related genes such as luxR (found two LuxR candidates in H. somni), narQ, uspE and uspA. Our group has reported that UspE is a global regulator controlling virulence and biofilm formation in H. somni. Finally, three of those sRNA candidates that interact with the above-mentioned biofilm/virulence genes were selected for further studies. Our initial plan was to overexpress two of the sRNAs in H. somni and characterize them by conducting phenotypic and genotypic studies. However, this approach did not work. Therefore, we concentrated on two sRNA candidates for the later part of experiments. We have modified the text.

Line 369: “…in the Northern blots indicated multiple transcripts…” replace “indicated” with “suggests”

We have changed it.

Line 363, Fig 4, and throughout the manuscript: When referring to the size of ssRNA molecules it is more accurate to refer to size in “bases” or “nucleotides (nt)” rather than “base pairs (bp)”.

We have corrected it as ‘bases’ throughout the manuscript and in figures.

Line 378: Target predictions using IntaRNA (or other target prediction software) are notoriously unreliable. Unless accompanied by experimental validation, these analyses are merely suggestive. Furthermore the details regarding how the IntaRNA analysis was performed are unclear. Consequently, I suggest removing section 3.6 from the results section. Some of the more interesting predicted interactions could be mentioned in the discussion section.

The predicted sRNA fragments were aligned using IntaRNA with the default settings. IntaRNA predicts interacting regions between two RNA molecules by incorporating the accessibility of both interaction sites and the presence of a seed interaction. We deleted the section 3.6 and moved some of the interesting, predicted interactions to the discussion section.

Line 401: Why were only 2 sRNAs investigated?

The reason why we selected the two specific sRNAs for investigation is that they could interact with known virulence-related genes in H. somni. These two were the most-promising candidates.

Line 410: What is the significance of the nucleotide residues in bold highlighted with an arrow in figure 6B and figure 6C?

The nucleotides in bold represent additional transcriptional start sites upstream of the transcriptional start site identified by RLM-RACE, which is indicated by the ‘arrow’. We have included it in the figure legend 4 (new figure no.).

Line 419: In the EMSA experiments shown in Fig 7, why are two bands of differing sizes detected for the in vitro transcribed RNA probes in all three experiments? Were the in vitro transcribed RNAs purified and DNAse treated? How was removal of DNA confirmed? Also it would be useful to include labels indicating the size and identity of each RNA molecule in Fig 7.

Yes, the in vitro transcribed probes were DNase treated, and the removal of DNA was confirmed by gel electrophoresis. Of the two bands of differing sizes detected in the in vitro transcribed RNA probes, the upper one is due to secondary structure formation. The reference figure indicating the secondary structure formation is given below. (Ref. Cheunim T et al. 2008, Virus Research 131(2-5):189-98).

We have indicated the size and identity of each RNA free probe in Fig 5 (new figure no.).

Line 431: “Once the 5’ and 3’ termini of the sRNA transcripts were assigned by RLM-RACE…” Identification of 3’ ends by RACE was not carried out. Please clarify this statement and/or add the 3’ RACE data.

We have not done 3’-RACE to identify the 3’ end of sRNAs. We have modified the sentence in the text.

Fig 8: The figure provided is extremely low resolution and blurry. Please provide high quality figures. Also the two individual RNAs should be labeled in the diagram.

We have modified Fig. 6 (new figure no.) and labelled the individual sRNAs.

Line 436: “The complex RNA-fold structures containing several stem loops in conjunction with the multiple target genes predicted by IntaRNA [40] is indicative of the potential of these sRNAs to control diverse biological processes.” This statement is a bit of an overreach and should be removed.

We have deleted the above-mentioned sentence in the revised manuscript.

Line 474: “The secondary structure (containing AU-rich regions and several stem loops) of the selected sRNAs confirms…” Please change the word “confirms” as the predicted secondary structures by themselves do not conclusively demonstrate anything.

We have changed “confirmed” to “indicated”.

Line 484 capitalize the G in gram negative. Same thing on line 494

We have modified it.

Line 485 “including pH”? Which pH are they referring to?

Technically, gram-negative does not refer to the inventor, Dr. Gram, as Gram strain does. Nonetheless, we have capitalized Gram everywhere as suggested.

Lines 480 to 505: This entire section of the discussion seems superfluous. It does not discuss any of the results generated in this manuscript, rather it discusses the roles of some small RNA's in other bacteria. It is unclear why this is included or if it is necessary.

We were trying to suggest the functional aspects of the sRNAs identified in this study by discussing that of their homologues well-studied in other bacteria. We have modified this section and made it more concise.

The authors should provide a more in-depth discussion of the similarities and differences between the sRNAs identified in this study and the study of Kumar et al (reference 8). For example, were the growth conditions similar? Can they speculate why certain sRNAs were identified in one study over another?

The differences between the sRNAs identified in our study with that of Kumar et al. could be due to the difference in culture conditions for the RNA extraction and the RNA type used for sequencing. In the study by Kumar et al., the H. somni 2336 cells were grown in Trypticase Soy Agar (TSA)-blood agar plates (with 5% sheep red blood cells) overnight, and the cells were washed with brain heart infusion (BHI) broth followed by total RNA isolation. Then, the mRNA enrichment was carried out using MICROBExpress. This enriched mRNA was sequenced, mapped, and intergenic regions were identified. The intergenic regions were analyzed to identify novel small RNAs. In our study, planktonic cultures of H. somni 2336 were cultured in Columbia broth (BD DifcoTM) with required growth supplements until stationary phase (reportedly higher production of Hfq). The cell extract was prepared and interacted with anti-Hfq IgG tagged magnetic beads to isolate Hfq and associated sRNAs. The purified sRNAs were directly used for sequencing followed by mapping and analysis to identify intergenic regions. Thus, in our study, we focused only on Hfq-associated sRNAs. However, the study by Kumar et al. reported the whole small RNA profile including the Hfq-associated sRNAs in H. somni. The data from Kumar et al., identified a number of sRNAs that were not detected in our study. In addition, there are sRNAs found in our study that were not detected by Kumar et al. The sRNAs isrK, tmRNA, gcvB, and those involved in the metabolism of lysine and glycine were identified in both studies, indicating their role in growth under both the culture conditions. We have modified the discussion section of the revised manuscript to indicate these differences.

We have used PACE to ensure that the figures in the revised manuscript meet PLOS requirements.

---

## [Decision Letter · Decision Letter 1]

20 Mar 2023

PONE-D-22-34191R1Detection and initial characterization of Hfq-associated sRNAs in *Histophilus somni* strain 2336*PLOS ONE*

Dear Dr. Inzana,

As you can see from their evaluations below, one reviewer was satisfied with your revised manuscript, but the other one still has some concerns that they feel were not adequately addressed. In looking at their concerns, I think they are justified. Consequently, I am going to ask you to submit a revised version of  the paper that addresses these concerns or provide a rebuttal or rebuttals stating why the concerns do not need to be addressed.

Thanks in advance for your patience with the process!  Both reviewers expressed the opinion that the work will be useful to the field, so I'd like to get the best possible version of the paper that we can into the literature.

Sincerely,

Marty Roop

Academic Editor

*PLOS ONE*

Journal Requirements:

Reviewers' comments:

Reviewer's Responses to Questions

**Comments to the Author**

1. If the authors have adequately addressed your comments raised in a previous round of review and you feel that this manuscript is now acceptable for publication, you may indicate that here to bypass the “Comments to the Author” section, enter your conflict of interest statement in the “Confidential to Editor” section, and submit your "Accept" recommendation.

Reviewer #1: All comments have been addressed

Reviewer #2: (No Response)

2. Is the manuscript technically sound, and do the data support the conclusions?

Reviewer #1: Yes

Reviewer #2: Yes

3. Has the statistical analysis been performed appropriately and rigorously? 

Reviewer #1: Yes

Reviewer #2: Yes

4. Have the authors made all data underlying the findings in their manuscript fully available?

Reviewer #1: Yes

Reviewer #2: No

5. Is the manuscript presented in an intelligible fashion and written in standard English?

Reviewer #1: Yes

Reviewer #2: Yes

6. Review Comments to the Author

Reviewer #1: Thank you for your corrections. You think that you misunderstand my comment about the 'title'. This was specifically for Fig 1 (that has been removed). For me the origional title of the manuscript was good.

Reviewer #2: This is my second time reviewing this manuscript. While most of my previous comments have been sufficiently addressed some have not. There are also a number of outstanding issues with the manuscript that need to be addressed before it can be considered for publication.

Issues

Line 56: “provide improved understanding of the differences…”

Line 221: “the miRNA…” should this be mRNA or RNA?

Line 245. EMSA's were carried out for three sRNAs however in the materials and methods section no oligos are given for the in vitro transcription reaction using HS9

Line 265 says six independent sRNA preparations were used in deep sequencing reactions but it is still not clear if these samples were individually barcoded and sequenced or if they were pooled and sequenced as one sample. The RNAseq data deposited online appears to only contain one data set so it would appear that all six samples were pooled and treated as one. This point was made in my original review and in response a table was included in the response to reviewers comments document showing the number of reads in each of the 6 samples. This table should be included in the manuscript. There is still a large discrepancy regarding the number of reads generated (see below) and the fact that the deposited data appears to be one dataset.

Line 271 says 20 million reads were generated however the RNAseq data deposited online contains only 2.4 million reads? The table provided in the response to reviewers comments also appaear to contain less than 2.4 million reads?

Fig 1. The figure legend for figure one is still incomplete. Line 300 says “the six black lines below it represent the six frame translation of the reference sequence” yet each black line is shown as one continuous line. What is the six frame translation of the reference sequence the authors refer to? The description of other six frame translation (at the top of the figures, line 298-299) is also unclear. I made this point in my original review and it was not addressed sufficiently. What does the orange line represent? For HS97 which of the three regions that have transcript mapping to them represents the small RNA?

Line 341-342 (and elsewhere) the authors refer to checking the relative expression level of the candidate sRNA's based on the sequencing data however no RNAseq whole transcriptome sequencing data is presented in the manuscript. What data are they referring to? This data should be made available and deposited online

Line 349 “Multi sequence analysis indicated that many of these sRNAs may interact with the Sigma 54 transcription factor”. Are the authors suggesting that these sRNAs interact with the messenger RNA encoding Sigma 54 or are they suggesting that the Sigma 54 protein directly binds to the SRNAs? Based on the data presented in table 3 it would appear that the latter is the case i.e they suggest that Sigma 54 protein binds to the RNAs. Are the authors suggesting that the Sigma 54 consensus binding site for DNA would be the same for single stranded RNA molecules? Is there any evidence for this?

Line 364 states that the selected eight sRNA candidates were chosen based on their predicted interaction with known biofilm and virulence related genes. However on line 343 it states that the 8 sRNA candidates were selected based on preliminary bioinformatic analysis and the fact that they were highly expressed in the sequencing data. Can the authors resolve this inconsistency?

Like 366 “based on their interaction with known virulence or biofilm genes” ... Are the authors referring to experimentally demonstrated interactions or bioinformatically predicted interactions. No reference is given?

Line 364 to 368 the rationale for going from 8 candidates to three candidates to two candidates is poorly described. No reason is given for the reduction from three to two candidate sRNAs.

Line 368 clearly states that the two most promising candidates (HS79 and HS97) were selected for further analysis…including northern blotting… yet then immediately on line 371 it's states that northern blocks were also performed for the third candidate HS9 (and included in Fig 3). Furthermore in the materials and methods section (Table 1) no sequence is given for the probe used to detect HS9 by northern blot.

In Figure 4 please indicate in panels B&C which one refers to HS79 and which refers to HS97

Line 486 and line 489: gcvB/GcvB is formatted differently. Please be consistent.

Line 496 “Target predictions for the select sRNAs”... Which select RNAs?

Line 503-504 should read “in addition some sRNAs were predicted to interact with their target genes...”

Line 514 should read “further studies on the sRNAs that are predicted to be associated with these…”

Line 530 “many of the sRNA candidates could potentially interact with genes”

7. PLOS authors have the option to publish the peer review history of their article (what does this mean?). If published, this will include your full peer review and any attached files.

Reviewer #1: No

Reviewer #2: No

---

## [Author Response · Author response to Decision Letter 1]

3 May 2023

Reviewer #1: Thank you for your corrections. You think that you misunderstand my comment about the 'title'. This was specifically for Fig 1 (that has been removed). For me the original title of the manuscript was good.

We have changed the title back to the original title in this revised manuscript.

Reviewer #2: This is my second time reviewing this manuscript. While most of my previous comments have been sufficiently addressed some have not. There are also a number of outstanding issues with the manuscript that need to be addressed before it can be considered for publication.

Issues

Line 56: “provide improved understanding of the differences…”

We have corrected it.

Line 221: “the miRNA…” should this be mRNA or RNA?

We have corrected it as RNA.

Line 245. EMSA's were carried out for three sRNAs however in the materials and methods section no oligos are given for the in vitro transcription reaction using HS9

We have included the oligo information for HS9 in Table 1.

Line 265 says six independent sRNA preparations were used in deep sequencing reactions, but it is still not clear if these samples were individually barcoded and sequenced or if they were pooled and sequenced as one sample. The RNAseq data deposited online appears to only contain one data set so it would appear that all six samples were pooled and treated as one. This point was made in my original review and in response a table was included in the response to reviewer’s comments document showing the number of reads in each of the 6 samples. This table should be included in the manuscript. There is still a large discrepancy regarding the number of reads generated (see below) and the fact that the deposited data appears to be one dataset.

We have included the RNA sequence summary as Table 2 in the revised manuscript. The data deposited online are the results from sample GSF2751-sRNA_Hfq-8. The discrepancy regarding the number of reads is addressed below in the next comment. We have modified the text in the revised manuscript as well.

Line 271 says 20 million reads were generated however the RNAseq data deposited online contains only 2.4 million reads? The table provided in the response to reviewer’s comments also appear to contain less than 2.4 million reads?

20 million reads is the combined total number of reads generated from all 6 sRNA samples. Sorry for this reporting error. We have modified the text accordingly in the revised manuscript. The table below (portion of Table 2 now in the text) shows the number of reads generated from each sRNA sample and their total (20.99 M). The RNAseq data deposited online (from sample GSF2751-sRNA_Hfq-8) contains 2.4 M reads. All 6 samples are repetitions of sequences from the same strain, so GSF2751-sRNA Hfq-8 is a representative sample. The last column in Table 2 in the revised manuscript is the number of mapped reads. 

Sample Barcode sequence PF Clusters 

GSF2751-sRNA_Hfq-1 CTATACAT 4,416,420 

GSF2751-sRNA_Hfq-11 TCGGCAAT 2,667,639 

GSF2751-sRNA_Hfq-3 CTCAGAAT 5,000,014 

GSF2751-sRNA_Hfq-7 GACGACAT 3,411,220 

GSF2751-sRNA_Hfq-8 TAATCGAT 2,413,457 

GSF2751-sRNA_Hfq-9 TCGAAGAT 3,090,434 

 Total: 20,999,184 

Fig 1. The figure legend for figure one is still incomplete. Line 300 says “the six black lines below it represents the six frame translation of the reference sequence” yet each black line is shown as one continuous line. What is the six-frame translation of the reference sequence the authors refer to? The description of other six frame translation (at the top of the figures, line 298-299) is also unclear. I made this point in my original review and it was not addressed sufficiently. What does the orange line represent? For HS97 which of the three regions that have transcript mapping to them represents the small RNA?

Below is the zoomed image of the six-frame translation (forward three frames and reverse three frames) of the H. somni genome reference sequence. The images in Fig. 1 was zoomed in to include more features including the whole sRNA regions which lead to the six frame translations showing as black lines. 

The six-frame translation of the reference sequence refers to the translation of the H. somni 2336 genome in three overlapping reading frames in the forward direction and the complementary strand in the reverse direction. 

Below is the zoomed image of the six-frame translation (forward three frames and reverse three frames) of the consensus sequence. The consensus sequence shows which residues are conserved (are always the same) and which residues are variable once we align the sequences to the reference genome sequence. The consensus is constructed from the most frequent residues at each site in the alignment.

If needed, we could remove the six-frame translation details of the reference genome sequence and the consensus sequence and include a revised version of Fig. 1.

The orange lines in Fig. 1 represent the maximum coverage for the RNA sequence. For HS97, the first region that has the transcript mapping represents the sRNA.

We have modified the Fig. 1 legend and the representative zoomed images of the six-frame translation of reference sequence and consensus sequence are shown as supporting information (S2 Supporting Information).

Line 341-342 (and elsewhere) the authors refer to checking the relative expression level of the candidate sRNA's based on the sequencing data however no RNAseq whole transcriptome sequencing data is presented in the manuscript. What data are they referring to? This data should be made available and deposited online.

The sRNA candidates were manually checked for their relative read coverage level in the sequencing data. The sRNAs with high read coverage were included for further analysis. We have modified the sentences in the revised manuscript.

Line 349 “Multi sequence analysis indicated that many of these sRNAs may interact with the Sigma 54 transcription factor”. Are the authors suggesting that these sRNAs interact with the messenger RNA encoding Sigma 54 or are they suggesting that the Sigma 54 protein directly binds to the SRNAs? Based on the data presented in table 3 it would appear that the latter is the case i.e they suggest that Sigma 54 protein binds to the RNAs. Are the authors suggesting that the Sigma 54 consensus binding site for DNA would be the same for single stranded RNA molecules? Is there any evidence for this?

Yes, we propose that the Sigma 54 protein binds to the sRNA regions in the genomic DNA, and not binding directly to the sRNAs. We have modified the Table 3 title and text in the revised manuscript.

Line 364 states that the selected eight sRNA candidates were chosen based on their predicted interaction with known biofilm and virulence related genes. However, on line 343 it states that the 8 sRNA candidates were selected based on preliminary bioinformatic analysis and the fact that they were highly expressed in the sequencing data. Can the authors resolve this inconsistency?

We have modified the text in the revised manuscript under “Northern blotting to verify selected sRNA candidates” to clarify that the selected sRNA candidates were chosen based on preliminary bioinformatic analysis and the relatively higher read coverage in the sequencing data.

Like 366 “based on their interaction with known virulence or biofilm genes” ... Are the authors referring to experimentally demonstrated interactions or bioinformatically predicted interactions. No reference is given?

Referring to bioinformatics predictions. We have modified the sentence in the revised manuscript.

Line 364 to 368 the rationale for going from 8 candidates to three candidates to two candidates is poorly described. No reason is given for the reduction from three to two candidate sRNAs.

Eight candidates were selected based on preliminary bioinformatics analysis and their relatively higher read coverage in the sequencing data. Out of the eight, three candidates were further selected based on their predicted interactions with known biofilm or virulence genes in H. somni. Our initial goal was to over-express and further characterize two of the most-promising sRNAs, and that’s why we selected only two sRNAs in the later part of our studies including determination of the transcriptional start site, prediction of secondary structures of selected sRNAs etc. We have modified the text in the revised manuscript and deleted the sentence “The two most promising candidates (HS79 and HS97) were selected for further analysis.”

Line 368 clearly states that the two most promising candidates (HS79 and HS97) were selected for further analysis…including northern blotting… yet then immediately on line 371 it's states that northern blocks were also performed for the third candidate HS9 (and included in Fig 3). Furthermore, in the materials and methods section (Table 1) no sequence is given for the probe used to detect HS9 by northern blot.

We have modified the text in the revised manuscript. The sentence stating that two of the most promising candidates (HS79 and HS97) were selected for further analysis under ‘Northern blotting to verify selected sRNA candidates’ has been deleted.. We have added the oligo details for HS9 for Northern blotting to Table 1.

In Figure 4 please indicate in panels B&C which one refers to HS79 and which refers to HS97

We have modified Fig. 4 to indicate the panels for HS79 and HS97.

Line 486 and line 489: gcvB/GcvB is formatted differently. Please be consistent.

We have corrected it in the revised manuscript.

Line 496 “Target predictions for the select sRNAs”... Which select RNAs?

Target predictions were carried out for a number of sRNAs including the 8 sRNAs with relatively high read coverage in the sequencing data. We have modified the sentence in the revised manuscript.

Line 503-504 should read “in addition some sRNAs were predicted to interact with their target genes...”

We have modified the sentence in the revised manuscript.

Line 514 should read “further studies on the sRNAs that are predicted to be associated with these…”

We have modified the sentence in the revised manuscript.

Line 530 “many of the sRNA candidates could potentially interact with genes”

We have modified the sentence in the revised manuscript.

---

## [Editor Report · Decision Letter 2]

10 May 2023

Detection and initial characterization of Hfq-associated sRNAs in *Histophilus somni *strain 2336

PONE-D-22-34191R2

Dear Tom,

Thanks for addressing the reviewers' concerns thoroughly and appropriately! Your manuscript is now considered scientifically suitable for publication and will be formally accepted for publication once it meets all outstanding technical requirements.

Thanks again, and I apologize for the delay in getting this decision to you!

Marty Roop

Academic Editor

*PLOS ONE*
---

## [Editor Report · Acceptance letter]

15 May 2023

PONE-D-22-34191R2 

Identification and initial characterization of Hfq-associated sRNAs in *Histophilus somni* strain 2336 

Dear Dr. Inzana:

I'm pleased to inform you that your manuscript has been deemed suitable for publication in PLOS ONE. Congratulations! Your manuscript is now with our production department. 

Kind regards, 

on behalf of

Dr. Roy Martin Roop II 

Academic Editor

PLOS ONE